# Important roles of the human leukocyte antigen class I and II molecules and their associated genes in the autoimmune coagulation factor XIII deficiency via whole-exome sequencing analysis

Tsukasa Osaki[1,2,3]*, Masayoshi Souri[1,2,3], Akitada Ichinose[1,2]

1 Japanese Collaborative Research Group on Autoimmune Coagulation Factor Deficiencies (JCRG supported by the Japanese Ministry of Health, Labor and Welfare), Yamagata, Japan, 2 Department of Molecular Patho-Biochemistry and Patho-Biology, Yamagata University School of Medicine, Yamagata, Japan, 3 Department of Public Health and Hygiene, Yamagata University Faculty of Medicine, Yamagata, Japan

* tosaki@med.id.yamagata-u.ac.jp

## Abstract

Autoimmune coagulation factor XIII deficiency is a bleeding disorder caused by the formation of autoantibodies against the coagulation factor XIII (FXIII); however, the molecular mechanism underlying this process is unknown. Therefore, in the present study, we aimed to elucidate this mechanism by performing whole-exome sequencing analysis of 20 cases of autoimmune FXIII deficiency. We identified approximately 21,788–23,916 variants in each case. In addition to their ability to activate T cells, present antigens, and immune tolerance, the candidate alleles were further narrowed down according to their allelic frequencies and the magnitude of damage caused by the substitution of amino acids. After selecting 44 candidate alleles, we investigated whether they were associated with the FXIII inhibitory titers and/or the anti-FXIII autoantibodies. We found that two polymorphisms whose variant allele frequencies were significantly lower in the patients tended to decrease FXIII inhibitory titers as the number of variant alleles increased. We also found that five polymorphisms whose variant allele frequencies were significantly higher in the patients tended to increase the levels of the anti-FXIII autoantibodies as the number of variant alleles increased. All of these polymorphisms were found in the human leukocyte antigen (HLA) class I and II molecules and their associated genes. In particular, the HLA class II molecule and its associated genes were found to be involved in the presentation of foreign antigens as well as the negative regulation of the proliferation of T-cells and the release of cytokines. Polymorphisms in the HLA class II molecules and the cytotoxic T lymphocyte antigen 4 have been reported to be associated with the development of autoantibodies in acquired hemophilia A. Therefore, we hypothesized that these polymorphisms may be associated with the development of autoantibodies in autoimmune FXIII deficiency.

**Data Availability Statement:** All relevant data are within the manuscript and its Supporting Information files.

**Funding:** Japan Society for the Promotion of Science, 16K09820, Akitada Ichinose Health and Labor Sciences Research Grant, 201911070A, Akitada Ichinose.

**Competing interests:** The authors have declared that no competing interests exist.

**Abbreviations:** FXIII, coagulation factor XIII; FXIII-A, coagulation factor XIII A subunit; FXIII-B, coagulation factor XIII B subunit; AH13, autoimmune hemorrhaphilia due to anti-FXIII/13 autoantibodies; AAXIII/13D, autoimmune acquired factor XIII/13 deficiency; AHA, acquired hemophilia A; HLA, human leukocyte antigen; SNP, single nucleotide polymorphism; CTLA-4, cytotoxic T-lymphocyte antigen 4; CD, cluster of differentiation; WES, whole-exome sequencing; GRCh37, HG19 Genome Reference Consortium Human Build 37; TMAP, Torrent Mapping Alignment Program; dbSNP, Single Nucleotide Polymorphism Database; ALFA, Allele Frequency Aggregator; ExAC, Exome Aggregation Consortium; GenomeAD, genome aggregation database; BU, Bethesda unit; ELISA, enzyme-linked immunosorbent assay; ICT, immunochromatographic test; AU, arbitrary unit; OR, odds ratio; MNP, multiple nucleotide polymorphism; AA, amino acid; GO, Gene Ontology; MHC, major histocompatibility complex; JCRG, Japanese Collaborative Research Group; MEXT, Japanese Ministry of Education, Culture, Sports, Science and Technology; MHLW, Japanese Ministry of Health, Labour and Welfare.

## Introduction

Coagulation factor XIII (FXIII) is a plasma pro-transglutaminase consisting of two catalytic A subunits (FXIII-A) and two carrier B subunits (FXIII-B). It plays an important role in maintaining hemostasis by cross-linking and stabilizing fibrin clots and increasing the resistance to mechanical stress and fibrinolysis [1,2]. FXIII deficiency results in severe bleeding diathesis, with the affected patients often requiring lifelong replacement therapy. The clinical symptoms of congenital and acquired FXIII deficiencies are very similar, ranging from multiple cutaneous mucosal bleeding to fatal intracavitary hemorrhage.

Acquired FXIII deficiency can either be an autoimmune or non-autoimmune disorder. Autoimmune FXIII deficiency is a rare autoimmune hemorrhagic disease [3–8], formerly known as autoimmune hemorrhaphilia due to anti-FXIII autoantibodies (AH13) [3] or autoimmune acquired factor XIII/13 deficiency (AAXIII/13D) [4]. Over the years, the incidence rate of autoimmune FXIII deficiency has been increasing in Japan; growing from 8 cases before 2000 to 51 cases in 2017 [3]. About half of the autoimmune FXIII deficiency cases are idiopathic in nature, while the other half are associated with underlying diseases [4]; however, all of them occur as a result of the spontaneous production of autoantibodies against endogenous FXIII. Autoimmune FXIII deficiency is characterized by a sudden onset of bleeding, which is often life-threatening, in patients with no history of bleeding, without either prolonged prothrombin time or prolonged activated partial thromboplastin time.

Approximately 50% of the total cases of acquired hemophilia A (AHA), are considered to be idiopathic [9]. AHA is associated with high frequencies of the human leukocyte antigen (HLA) class II alleles and single nucleotide polymorphisms (SNPs) of the cytotoxic T-lymphocyte antigen 4 (*CTLA-4*) gene [9–13]. These genetic factors are also associated with the development of factor VIII inhibitors in patients with severe hemophilia A [14–16]. HLA class II alleles play essential roles in the presentation of factor VIII peptides to the cluster of differentiation (CD)-4+ T-lymphocytes, while CTLA-4 acts as a negative regulator of the activation of T-cells. The variants of these genes are thought to be associated with the development of AHA in combination with other genetic and/or environmental factors.

Autoimmune FXIII deficiency, like other autoimmune diseases, is thought to be caused by the disruption of the immune system. A combination of genetic and environmental factors can impair immune tolerance, leading to the development of autoantibodies against FXIII along with the ageing of the immune system in elderly individuals. Nearly 47% of the patients with autoimmune FXIII deficiency have underlying diseases, including other autoimmune diseases (17%), diabetes (9%), and cancer (6%), while the rest of the cases, with no underlying diseases are classified as idiopathic [4]. The genetic factors associated with autoimmune FXIII deficiency have not yet been identified and the etiology of autoimmune FXIII deficiency also remains unknown.

To identify the genetic factors at risk of producing the FXIII inhibitors in patients with autoimmune FXIII deficiency, we performed whole-exome sequencing (WES) analysis of 20 autoimmune FXIII deficiency cases and investigated whether polymorphisms were associated with FXIII inhibitory titers and levels of anti-FXIII autoantibodies in these patients.

## Materials and methods

### Materials

Recombinant FXIII-A was kindly provided by Zymogenetics (Seattle, WA, USA). Anti-FXIII-A monoclonal antibody was obtained from Prof. Reed (Massachusetts General Hospital, Boston, MA, USA). Peroxidase-conjugated anti-human IgG antibodies were purchased from

MP Biomedicals (Solon, OH, USA). Tetramethylbenzidine peroxidase substrate kits were purchased from Bio-Rad Laboratories (Hercules, CA, USA).

## Clinical samples

We were consulted by physicians from all over Japan, from Hokkaido in the north to Okinawa in the south, in charge of the patients with unexplained hemorrhage. For this study, we recruited patients with severe bleeding who did not present any personal or family history of bleeding, from June 2003 to Aug 2016. A total of 48 cases of autoimmune FXIII deficiency with the FXIII activities measured using the amine incorporation assay [17] below the standard value of 0.7 IU/mL and with anti-FXIII autoantibodies [4] were included in this study. Of these, we collected the peripheral blood cells from 31. This study was approved by the Institutional Review Board of Yamagata University School of Medicine. All procedures were conducted in accordance with the Declaration of Helsinki. Written informed consent was obtained from all individuals.

The data about the Japanese population was obtained from "Population by Sex and Sex ratio for Prefectures—Total population, Japanese population, October 1, 2016" on the official statistics portal site of Japan (https://www.e-stat.go.jp/en/stat-search/files?page=1&layout=datalist&toukei=00200524&tstat=000000090001&cycle=7&year=20160&tclass1=000001011679&tclass2val=0).

## NGS library and template preparation

Genomic DNA was extracted from the peripheral blood cells of each of the 31 patients using standard phenol/chloroform methods [18]. The lengths of the DNA fragments were measured by capillary electrophoresis using a 2200 Tape Station Instrument with a High Sensitivity D1000 Screen Tape and Reagents (Agilent Technologies Japan, Ltd., Tokyo, Japan) and the DNA integrity number was calculated [ranging from 1 (highly degraded genomic DNA) to 10 (intact genomic DNA)]. As DNA with low integrity does not provide sufficient information in WES, we excluded the samples having a DNA integrity number less than 5. Finally, 20 samples were selected and next-generation amplicon-based sequencing (Ion Proton™ System; Life Technology Japan Ltd., Tokyo, Japan) was performed. DNA samples were amplified using premixed AmpliSeq primer pools and an Ion AmpliSeq HiFi mix (Ion AmpliSeq Library Kit v2.0; Life Technology Japan Ltd.). The resulting multiplex amplicons were treated with the FuPa reagent (Life Technology Japan Ltd.) to partially digest the primer sequences and phosphorylate the amplicons. Then, the amplicons were ligated to the Ion Xpress barcode adapters (Life Technology Japan Ltd.) according to the manufacturer's instructions. Library quantification was performed using a 2200 Tape Station Instrument with a High Sensitivity D1000 Screen Tape and Reagents. The amplified library was subjected to an emulsion polymerase chain reaction using the Ion OneTouch™ 2 Instrument with the Ion PI™ Template OT2 200 Kit v3 (Life Technology Japan Ltd.). Ion sharing particles were concentrated using the Ion OneTouch ES (Life Technology Japan Ltd.) and loaded onto the Ion PI Sequencing 200 Kit v3 (Life Technology Japan Ltd.).

## Ion Torrent data analysis

Signal Processing, base calls, and barcode deconvolution were performed using the Torrent Suite™ Software v 5.0.2 (Life Technology Japan Ltd.) [19]. Alignment to the HG19 Genome Reference Consortium Human Build 37 (GRCh37) was performed using the Torrent Mapping Alignment Program (TMAP; Life Technology Japan Ltd.) and the alignment output was in the BAM format. Torrent Suite™ Software v 5.0.2 was also used to generate relevant execution

metrics, such as the total number of sequences per sample. Individual amplicon coverage metrics were calculated using the coverage Analysis plug-in in Torrent Suite™ Software. Variants were identified using the variant Caller plug-in in Torrent Suite™ Software. The output results were in the VCF format, and were filtered and selected if their coverage was 15 x or higher within the exome target region.

All variants were filtered based on the minor allelic frequencies ($< 0.01$) and their potential harmful effects to accurately identify the rare and harmful variants. Allelic frequencies were obtained from the Single Nucleotide Polymorphism Database (dbSNP) (https://www.ncbi. nlm.nih.gov/snp/), including several public databases, such as the Allele Frequency Aggregator (ALFA), 1000 Genomes, Exome Aggregation Consortium (ExAC), and the Genome Aggregation Database (GenomeAD)-Genomes and GenomeAD-Exomes databases. The pathogenicity of the missense changes was assessed using the following *in silico* predictions: PROVEAN (http://provean.jcvi.org/genome_submit_2.php) and PolyPhen-2 (http://genetics.bwh. harvard.edu/pph2/).

## Allele call thresholds

A simple designation for the alleles was used based on more stringent allelic frequency thresholds, with reference to a previously reported paper [19]. SNP amplicons with allelic frequencies $\geq 90\%$ were considered to be homozygous for that allele, while those with allelic frequencies between 10% and 90% were considered to be heterozygous. All amplicons with allelic frequencies $< 10\%$ were ignored.

## Quantification of the FXIII inhibitory titer by ammonia release assay

The plasma samples of the patients were diluted 2-fold with saline and incubated with equal volumes of standard human plasma (Sysmex Corporation, Kobe, Japan) at 37°C for 2 h. The FXIII activity of the reaction mixture was measured using the Berichrom® FXIII ammonia release assay (Sysmex Corporation) according to the manufacturer's instructions. One Bethesda unit (BU) is defined as the amount of inhibitor that results in a residual activity of 50% in the mixture.

## Detection of anti-FXIII-A autoantibodies using enzyme-linked immunosorbent assay (ELISA)

ELISA was performed to detect the anti-FXIII-A autoantibodies in the plasma samples of the patients, as previously described [17]. Patient plasma (0.5 μL) was diluted 10-fold with 20 mM Tris-buffered saline (pH 7.5) containing 2% bovine serum albumin and incubated with 100 ng of recombinant FXIII-A at 37°C for 1 h. The reaction mixture was further diluted 100-fold with the same buffer, pipetted into a 96-well plate coated with an anti-FXIII-A monoclonal antibody (100 ng), and incubated at 37°C for 1 h. The plate was incubated with peroxidase-conjugated anti-human IgG. Detection of the anti-FXIII-A autoantibodies bound to recombinant FXIII-A was performed using the tetramethylbenzidine substrate as previously described [17]. The relative absorbance of autoimmune FXIII deficiency-1 at 450 nm was set as 1.0.

## Detection of anti-FXIII-A autoantibodies using the immunochromatographic test (ICT)

ICT was performed to detect the anti-FXIII-A autoantibodies as previously described [20]. The plasma samples were diluted with saline and incubated with equal volumes of standard human plasma at 37°C for 5 min. Then, the anti-FXIII-A autoantibodies in the reaction

mixture were detected using the in-house anti-FXIII-A monoclonal antibodies applied to a nitrocellulose strip and anti-human Ig (G+M+A)-colloidal gold conjugate. The line intensity proportional to the amount of anti-FXIII-A autoantibodies visualized using colloidal gold was read using a reader device (Fact Scan; Denken Co., Ltd., Oita, Japan) and expressed as a unit of absorbance relative to the absorbance of the positive control plasma [assigned as 1 arbitrary unit (AU)].

## Statistical analysis

The variant allelic frequencies of the patients with autoimmune FXIII deficiency were calculated as follows: for chromosomes 1–22, [2*(number of cases with homozygous variant alleles) + (number of cases with heterozygotes)]/[2*(total number of cases)]; for chromosome X, [2* (number of female cases with homozygous variant alleles) + (number of female cases with heterozygotes) + (number of male cases with variant alleles)]/[2*(total number of female cases) + (total number of male cases)]; and for chromosome Y, [(number of male cases with variant alleles)/(total number of male cases)]. The frequency of the reference allele was calculated as: [1-(variant allelic frequency)]. The odds ratio (OR) was calculated as follows: [(ratio of variant allelic frequency to reference allelic frequency in all autoimmune FXIII deficiency cases)/(ratio of variant allelic frequency to reference allelic frequency registered in the database)]. In this study, we focused on polymorphisms with ORs < 0.67 or > 1.5. Comparisons of the autoimmune FXIII deficiency case distribution and relative allelic frequency risk were performed using a chi-square test or two-tailed Fisher's exact test in the JMP software v.12.2.0 (SAS Institute, Cray, NC, USA). Statistical significance was set at P < 0.05.

## Results

### WES analysis of autoimmune FXIII deficiency

We performed next-generation sequencing analysis of the whole-exome of 20 autoimmune FXIII deficiency cases from 19 institutions. The distribution of autoimmune FXIII deficiency was not significantly different in different areas of Japan; however, two cases were identified in the Gunma prefecture (S1 and S2 Tables). This study included 12 males and 8 females, aged 55–88 years with a median age of 75 (Table 1). Chromosome 1 had the highest number of variants, accounting for approximately 11% of the total variants (S1A Fig). Homozygous variants accounted for approximately 42%, while heterozygous variants accounted for the remaining 58% of all variants (S1B Fig). The SNP number in each patient was observed from 21,026– 23,037, with a median value of 22,415, which accounted for approximately 97% of the total variants (S1C Fig and Table 1). The number of multiple nucleotide polymorphisms (MNPs) was 153–195 with a median value of 176. The deletion number was 268–431 with a median value of 358, and the insertion number was 217–291 with a median value of 265. Approximately 87% of these variants were derived from exons, while the remaining 13% were derived from the introns close to exons. When classified on the basis of the amino acid (AA) mutations due to exon variants, approximately 52% of the cases involved 1-AA substitutions, while 46% exhibited synonymous mutations (S1D Fig). Approximately 9% of the exon variants were predicted to be damaged, while the remaining 91% were predicted to be tolerated.

### Selection of candidate alleles associated with the development of anti-FXIII autoantibodies in autoimmune FXIII deficiency

We selected candidate alleles in three ways (Fig 1). First, we focused on the variants of the exon regions in *F13A1*, *F13B*, *CTLA4*, *HLA-DRB1*, and *HLA-DQB1*. Second, we focused on

**Table 1. Summary of variants in 20 autoimmune FXIII deficiency cases.**

| Case No. | Sex | Age (yr) | FXIII activity (U/mL) | Underling disease | SNP | MNP | Del | Ins |
|---|---|---|---|---|---|---|---|---|
| 1 | M | 55 | 0.06 | None | 22,598 | 171 | 369 | 283 |
| 4 | M | 78 | 0.27 | Unknown | 22,382 | 184 | 347 | 291 |
| 6 | M | 79 | <0.02 | AAA | 22,896 | 185 | 364 | 285 |
| 8 | F | 76 | 0.26 | None | 22,708 | 176 | 397 | 280 |
| 9 | M | 75 | <0.02 | AAA + IP | 22,235 | 184 | 410 | 254 |
| 10 | F | 83 | 0.06 | DM | 22,830 | 171 | 318 | 266 |
| 11 | F | 78 | 0.09 | HT + DL | 21,150 | 153 | 268 | 217 |
| 15 | M | 63 | 0.15 | Unknown | 21,981 | 187 | 342 | 252 |
| 17 | F | 73 | 0.26 | None | 22,316 | 162 | 314 | 247 |
| 19 | M | 75 | 0.60 | None | 21,026 | 175 | 431 | 259 |
| 20 | M | 88 | 0.37 | DVT | 21,038 | 164 | 354 | 242 |
| 27 | M | 71 | 0.02 | BT | 23,037 | 195 | 402 | 282 |
| 28 | M | 70 | <0.02 | SLE + AIHA | 22,570 | 164 | 312 | 251 |
| 29 | M | 65 | 0.04 | Unknown | 21,659 | 158 | 311 | 238 |
| 35 | F | 77 | 0.04 | None | 22,360 | 167 | 328 | 264 |
| 39 | F | 68 | 0.18 | DL | 21,955 | 187 | 299 | 245 |
| 40 | F | 80 | 0.27 | DM + HT | 22,448 | 193 | 375 | 281 |
| 42 | M | 71 | <0.02 | None | 22,613 | 174 | 367 | 274 |
| 44 | F | 68 | 0.10 | RA | 22,591 | 180 | 389 | 268 |
| 48 | M | UNK | 0.62 | Unknown | 22,465 | 178 | 362 | 279 |

M and F indicate male and female, respectively. FXIII activity was measured using an ammine-incorporation assay. The abbreviations for underlying diseases are as follows: AAA; abdominal aortic aneurysm, IP; interstitial pneumonia; DM; diabetes mellitus, HT; hypertension, DL; dyslipidemia, DVT; deep venous thrombosis, BT; bladder tumor, SLE; systemic lupus erythematosus, AIHA; autoimmune hemolytic anemia, RA; rheumatoid arthritis. The number of SNPs, MNPs, deletions (Del), and insertions (Ins) is entered in the table.

variants that caused AA changes including frameshift mutations in the genes associated with the Gene Ontology (GO) terms "T cell activation," "antigen presentation," and/or "immune tolerance." Third, we selected the best candidate alleles to efficiently identify rare and damaging variants.

**Exon variants in *F13A1*, *F13B*, *CTLA4*, *HLA-DRB1*, and *HLA-DQB1* genes (Selection of Candidate 1).** We hypothesized that *F13A1* and *F13B* variants might also be associated with the development of autoantibodies in patients with autoimmune FXIII deficiency as *F8* is known to be associated with the development of autoantibodies in patients with AHA [13]. We identified four *F13A1* variants and three *F13B* variants. All the *F13A1* variants and one *F13B* variant caused a single AA change, while two of the four *F13B* variants caused a synonymous change. In these variants, one *F13A1* variant (rs5982, p.Pro565Leu) had an OR < 0.67 against datasets of all databases from the Asian or East Asian regions. Another of *F13A1* variant (rs76451285, p.Ala395Val) exhibited an OR > 1.5 except for ALFA database. The *F13B* variant (rs6003, p.Arg115His) was found to be homozygous in autoimmune FXIII deficiency; however, its allelic frequencies were found to be > 0.9 in all the databases.

We also investigated *CTLA4*, *HLA-DRB1*, and *HLA-DQB1*, whose variant allelic frequencies in patients with AHA differ from those in the control cohort [9–11]. We identified 2, 6, and 28 *CTLA4*, *HLA-DRB1*, and *HLA-DQB1* variants, respectively (Tables 2 and S3). In these variants, one *CTLA4* and two *HLA-DQB1* variants exhibited an OR > 1.5 against the datasets from all databases of Asia or East Asia, while two *HLA-DRB1* and six *HLA-DQB1* variants exhibited an OR < 0.67.

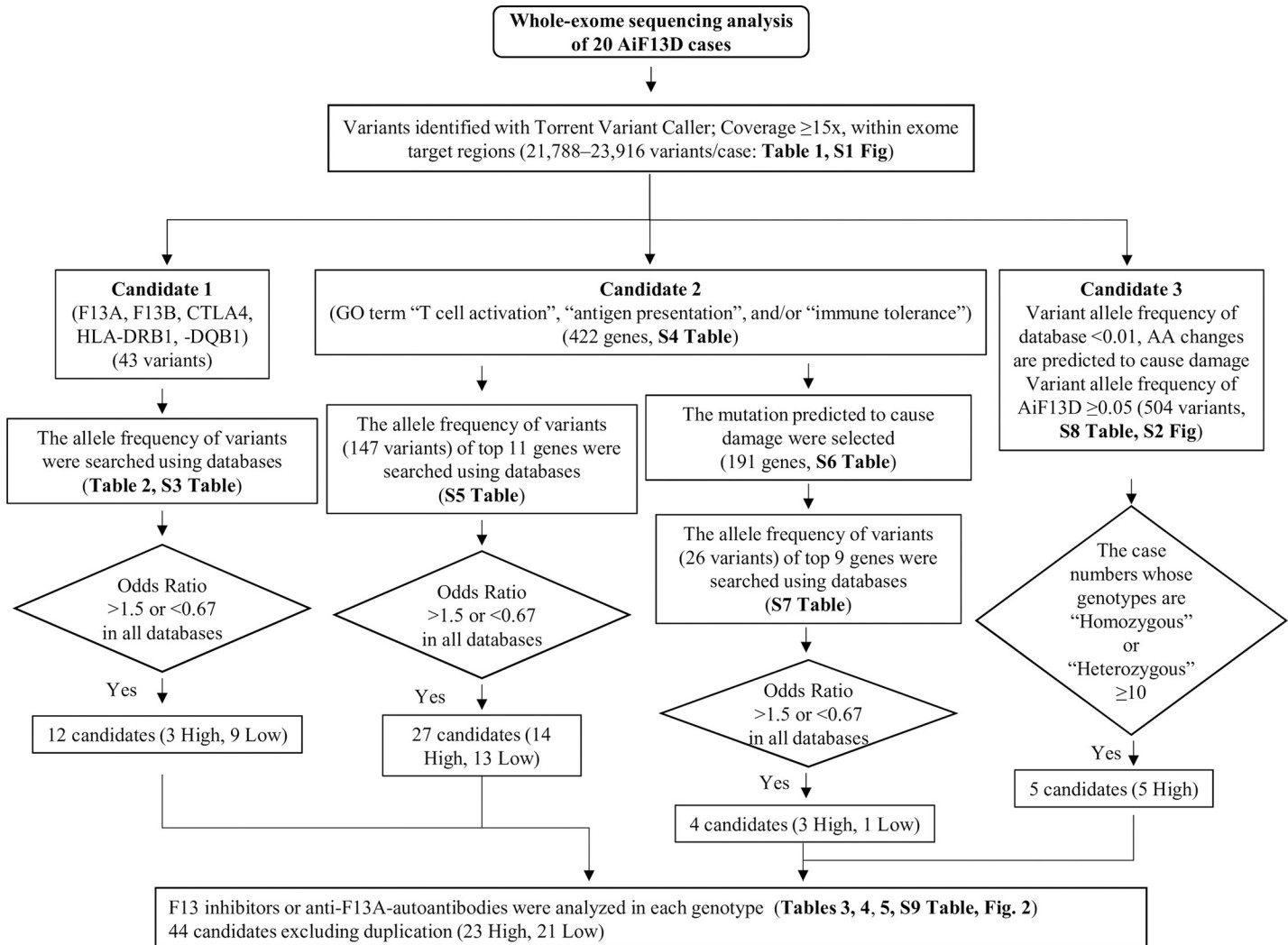

**Fig 1. Data analysis and selecting candidate alleles above the criteria from genetic variants identified in whole exome sequencing of 20 autoimmune FXIII deficiency cases.** The analytical dataset was obtained by merging all the annotated variants called in the 20 autoimmune FXIII deficiency cases into a single dataset. The candidate alleles were subsequently narrowed down by three selection methods.

**Damaged exon variants associated with GO terms "T cell activation," "antigen presentation," and "immune tolerance" (Selection of Candidate 2).** We investigated 818 genes, without duplication, including 677 genes associated with the GO term, "T cell activation," 233 with the GO term, "antigen presentation," and 26 with the GO term, "immune tolerance." A total of 7,055 variants, including 422 redundant genes, were identified (S4 Table). The top 11 genes, including *HLA-DQB1*, possessed > 100 variants (five variants/case). We summarized these 11 genes, including their ORs compared with the allelic frequencies registered in the databases (S5 Table). Among these variants, three *BTNL2* variants, two *SPINK5*, *HLA-C*, and *HLA-DPB1* variants, and one *HLA-A*, *HLA-B*, *MICA*, *HLA-DQB1*, and *SIRPA* variant exhibited ORs > 1.5 (compared with the allelic frequencies registered in the databases), while four *HLA-DPB1* variants, three *SPINK5* variants, two *HLA-B*, *MICA*, and *HLA-DQB1* variants exhibited ORs < 0.67.

We also narrowed down the candidate variants based on the damage caused by the change in nucleotides. We identified a total of 1,029 variants, including those in 191 redundant genes

**Table 2.** *F13A1, F13B, CTLA4, HLA-DRB1*, and *HLA-DQB1* variants in autoimmune FXIII deficiency cases.

| Chr | Pos | Ref | Var | Gene ID | Pos (AA) | Ref (AA) | Var (AA) | Type | Prediction (cutoff = 0.05) | dbSNP_ID | Cases | ExAc | |
|---|---|---|---|---|---|---|---|---|---|---|---|---|---|
| | | | | | | | | | | | | Asian | |
| | | | | | | | | | | | | Frequency | OR |
| 1 | 197009798 | A | G | *F13B* | 602 | N | N | Synonymous | Tolerated | rs5998 | 8.50E-01 | 6.71E-01 | **2.78** |
| 1 | 197030201 | T | C | *F13B* | 152 | T | T | Synonymous | Tolerated | rs5997 | 1.00E+00 | 9.35E-01 | ND |
| 1 | 197031021 | C | T | *F13B* | 115 | R | H | Single AA Change | Tolerated | rs6003 | 1.00E+00 | 9.29E-01 | ND |
| 2 | 204732714 | A | G | *CTLA4* | 17 | T | A | Single AA Change | Tolerated | rs231775 | 5.75E-01 | NA | ND |
| 2 | 204737478 | C | G | *CTLA4* | 169 | P | A | Single AA Change | Damaging | rs74808460 | 2.50E-02 | 2.00E-04 | **128.18** |
| 6 | 6152137 | C | G | *F13A1* | 652 | E | Q | Single AA Change | Tolerated | rs5988 | 1.00E-01 | 1.72E-01 | **0.54** |
| 6 | 6152140 | C | T | *F13A1* | 651 | V | I | Single AA Change | Tolerated | rs5987 | 1.00E-01 | 1.13E-01 | 0.87 |
| 6 | 6174866 | G | A | *F13A1* | 565 | P | L | Single AA Change | Tolerated | rs5982 | 2.00E-01 | 3.30E-01 | **0.51** |
| 6 | 6197488 | G | A | *F13A1* | 395 | A | V | Single AA Change | Tolerated | rs76451285 | 5.00E-02 | 3.70E-03 | **14.17** |
| 6 | 32548581 | A | G | *HLA-DRB1* | 235 | F | F | Synonymous | Tolerated | rs113175445 | 1.50E-01 | 2.95E-01 | **0.42** |
| 6 | 32549525 | C | G | *HLA-DRB1* | 154 | G | A | Single AA Change | Damaging | rs111965977 | 5.00E-02 | 1.90E-01 | **0.22** |
| 6 | 32549531 | T | C | *HLA-DRB1* | 152 | Y | C | Single AA Change | Damaging | rs112796209 | 5.00E-02 | 1.90E-01 | **0.22** |
| 6 | 32549596 | T | C | *HLA-DRB1* | 130 | V | V | Synonymous | Tolerated | | 1.25E-01 | NA | ND |
| 6 | 32549611 | T | C | *HLA-DRB1* | 125 | Q | Q | Synonymous | Tolerated | rs1071752 | 1.25E-01 | NA | ND |
| 6 | 32549613 | GG | CA | *HLA-DRB1* | 125 | Q | E | Single AA Change | Tolerated | | 1.25E-01 | NA | ND |
| 6 | 32629755 | G | A | *HLA-DQB1* | 217 | T | I | Single AA Change | Tolerated | rs1130399 | 3.25E-01 | 2.31E-01 | **1.60** |
| 6 | 32629764 | C | T | *HLA-DQB1* | 214 | S | N | Single AA Change | Tolerated | rs1130398 | 4.75E-01 | 3.89E-01 | 1.42 |
| 6 | 32629802 | A | G | *HLA-DQB1* | 201 | D | D | Synonymous | Tolerated | rs1049092 | 7.00E-01 | 6.97E-01 | 1.02 |
| 6 | 32629809 | C | T | *HLA-DQB1* | 199 | R | H | Single AA Change | Tolerated | rs701564 | 3.75E-01 | NA | ND |
| 6 | 32629847 | A | G | *HLA-DQB1* | 186 | T | T | Synonymous | Tolerated | rs1049133 | 8.00E-01 | 8.92E-01 | **0.49** |
| 6 | 32629859 | A | G | *HLA-DQB1* | 182 | N | N | Synonymous | Tolerated | rs1049130 | 8.00E-01 | 7.00E-01 | **1.71** |
| 6 | 32629868 | A | G | *HLA-DQB1* | 179 | L | L | Synonymous | Tolerated | rs1049088 | 1.00E-01 | 1.94E-01 | **0.46** |
| 6 | 32629889 | G | A | *HLA-DQB1* | 172 | A | A | Synonymous | Tolerated | rs1049087 | 5.50E-01 | 5.39E-01 | 1.05 |
| 6 | 32629891 | C | T | *HLA-DQB1* | 172 | A | T | Single AA Change | Tolerated | rs1063323 | 4.75E-01 | 3.61E-01 | **1.60** |
| 6 | 32629904 | A | G | *HLA-DQB1* | 167 | D | D | Synonymous | Tolerated | rs1049086 | 7.00E-01 | 6.94E-01 | 1.03 |
| 6 | 32629920 | C | T | *HLA-DQB1* | 162 | R | Q | Single AA Change | Tolerated | rs41544112 | 1.25E-01 | 3.01E-02 | **4.61** |
| 6 | 32629935 | C | G | *HLA-DQB1* | 157 | G | A | Single AA Change | Tolerated | rs1063322 | 4.75E-01 | 5.44E-01 | 0.76 |
| 6 | 32629936 | C | T | *HLA-DQB1* | 157 | G | S | Single AA Change | Tolerated | rs1049107 | 1.00E-01 | 2.02E-01 | **0.44** |
| 6 | 32629955 | C | T | *HLA-DQB1* | 150 | S | S | Synonymous | Tolerated | rs1063321 | 4.75E-01 | 3.41E-01 | **1.75** |
| 6 | 32629963 | C | T | *HLA-DQB1* | 148 | V | I | Single AA Change | Tolerated | rs1049100 | 1.00E-01 | 2.12E-01 | **0.41** |
| 6 | 32632744 | C | T | *HLA-DQB1* | 70 | A | A | Synonymous | Tolerated | rs1049082 | 3.25E-01 | 3.67E-01 | 0.83 |

(*Continued*)

**Table 2.** (Continued)

| Chr | Pos | Ref | Var | Gene ID | Pos (AA) | Ref (AA) | Var (AA) | Type | Prediction (cutoff = 0.05) | dbSNP_ID | Cases | ExAc Asian Frequency | OR |
|---|---|---|---|---|---|---|---|---|---|---|---|---|---|
| 6 | 32632745 | G | A | HLA-DQB1 | 70 | A | V | Single AA Change | Tolerated | rs1063318 | 2.75E-01 | 4.44E-01 | **0.47** |
| 6 | 32632749 | A | C | HLA-DQB1 | 69 | Y | D | Single AA Change | Tolerated | | 2.25E-01 | NA | ND |
| 6 | 32632770 | A | G | HLA-DQB1 | 62 | Y | H | Single AA Change | Tolerated | | 1.75E-01 | NA | ND |
| 6 | 32632777 | C | T | HLA-DQB1 | 59 | V | V | Synonymous | Tolerated | rs1049068 | 1.50E-01 | 1.02E-01 | **1.55** |
| 6 | 32632781 | AGA | TAA/CCC | HLA-DQB1 | 58 | L | Y/G | Single AA Change | Tolerated/Tolerated | | 3.50E-01/1.00E-01 | NA | ND |
| 6 | 32632790 | C | A | HLA-DQB1 | 55 | R | L | Single AA Change | Damaging | rs41540813 | 5.00E-02 | 1.94E-02 | **2.65** |
| 6 | 32632795 | C | G | HLA-DQB1 | 53 | T | T | Synonymous | Tolerated | rs1049079 | 7.50E-02 | NA | ND |
| 6 | 32632801 | G | A | HLA-DQB1 | 51 | N | N | Synonymous | Tolerated | rs3204373 | 2.25E-01 | 1.43E-01 | **1.74** |
| 6 | 32632818 | T | G | HLA-DQB1 | 46 | M | L | Single AA Change | Tolerated | rs1130368 | 2.50E-02 | 1.78E-01 | **0.12** |
| 6 | 32632820 | C | G | HLA-DQB1 | 45 | G | A | Single AA Change | Tolerated | rs1130375 | 3.75E-01 | NA | ND |
| 6 | 32632832 | A | T | HLA-DQB1 | 41 | F | Y | Single AA Change | Tolerated | rs9274407 | 5.25E-01 | 7.79E-01 | **0.31** |
| 6 | 32632833 | A | G | HLA-DQB1 | 41 | F | L | Single AA Change | Tolerated | rs12722107 | 2.25E-01 | NA | ND |

When the OR of autoimmune FXIII deficiency against Asia database in ExAc was > 1.5 or < 0.67, the OR was represented in bold letters. NA; not available because the variant frequency was not registered in the database. ND; not determined because the variant frequency was not registered in the Asian database. In addition, OR was not determined by following two; 1) the variant frequency registered in the database was 0.0, or 2) the variant frequency of autoimmune FXIII deficiency was 1.0.

(S6 Table). The top 9 genes, including *MICA* and *HLA-C*, possessed 20 or more variants (one variant/case). We also summarized these 9 genes, including their ORs and compared them with the allelic frequencies registered in the databases (S7 Table). In these variants, one variant each of *ITPKB*, *MICA*, and *PSMA7* exhibited ORs > 1.5 (compared with the allelic frequencies registered in the databases), while one variant of *P2RX7* had an OR < 0.67.

**Damaged exon variants with allelic frequencies less than 0.01 (Selection of Candidate 3).** We narrowed down candidate alleles based on the damage described above, exhibiting variant allelic frequencies < 0.01 in the databases, but ≥ 0.05 in autoimmune FXIII deficiency cases. As a result, 64 variants met the criteria, while 440 variants were not registered in the allelic frequency database (S8 Table). A total of 504 of these were selected as candidate alleles. With respect to the number of variants per chromosome, chromosome 3 had the largest number of variants, accounting for approximately 16% of the total variants (S2A Fig). Homozygous variants accounted for about 43%, while heterozygous variants accounted for the remaining 57% of the total variants (S2B Fig). The deletion number in each patient was observed from 30 to 68 with a median of 43, which occupied approximately 50% of the total (S2C Fig). The SNP number was 13–37 with a median of 27, the MNP number was 4–10 with a median of 6, and the insertion number was 4–15, with a median of 11. When classified by codon mutations due to exon variants, the frameshift mutations were found to be the highest, about 55% (S2D Fig). We further selected five candidate alleles based on 10 or more (half or more) cases with the terms, "heterozygous" or "homozygous."

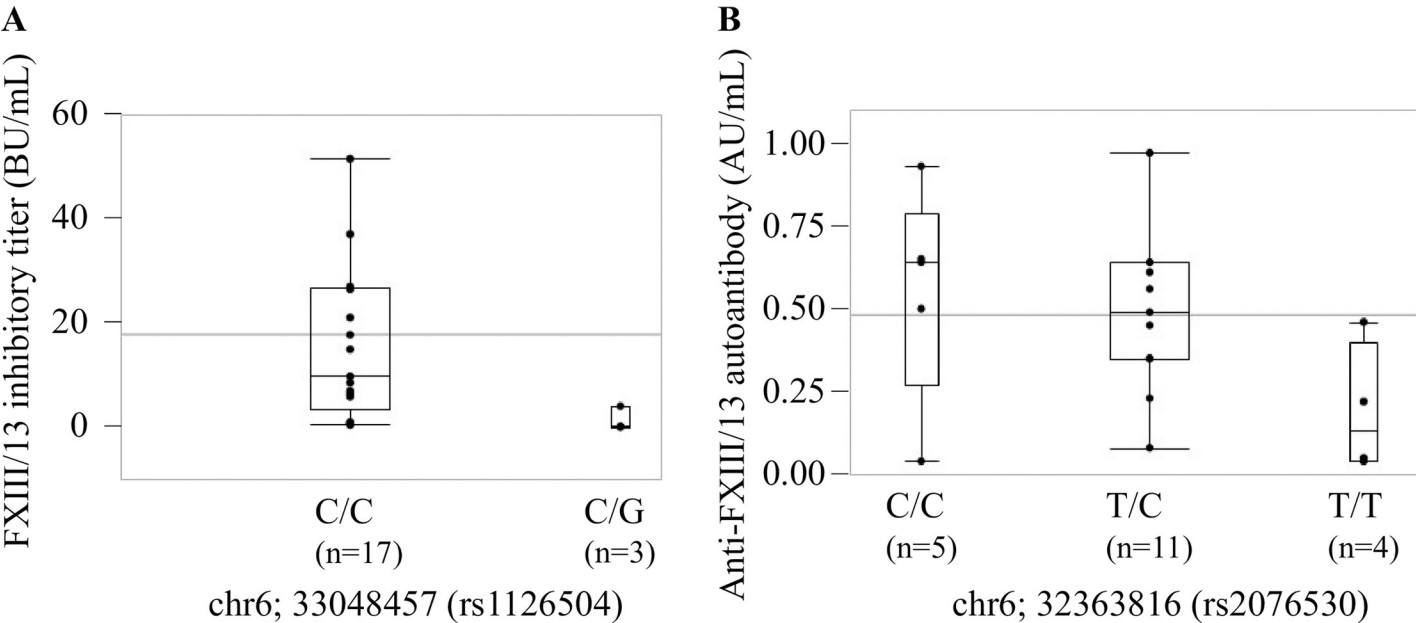

**Fig 2. FXIII inhibitory titers and anti-FXIII autoantibody levels in various variants.** *A*, FXIII inhibitory titers (Bethesda unit; BU) in reference allele (C/C) and heterozygous allele (C/G) at chromosome 6; 33048457 in *HLA-DPB1* (rs1126504). *B*, Anti-FXIII autoantibody levels measured using ICT (arbitrary unit; AU) as described in "*Materials and Methods*" in reference allele (T/T), heterozygous allele (T/C) and homozygous allele (C/C) at chromosome 6; 32363816 in *BTNL2* (rs2076530).

## Association of selected candidate alleles with FXIII inhibitory titers and/or levels of anti-FXIII-A autoantibodies

We then investigated the association of these variants with the FXIII inhibitory titers and/or levels of anti-FXIII-A autoantibodies. Twenty autoimmune FXIII deficiency cases were divided into three groups using each polymorphism as an index, i.e., cases with variant allele homozygotes, cases with heterozygotes, and cases with reference allele homozygotes. We compared the FXIII inhibitory titers and/or levels of anti-FXIII-A autoantibodies measured by ELISA or ICT in each genotype of 44 candidate alleles (Fig 2, Tables 3–5 and S9). Of these, 21 had significantly lower variant frequencies in autoimmune FXIII deficiency cases and 23 had significantly higher variant frequencies than those registered in the database. Among polymorphisms with significantly lower variant allele frequencies in autoimmune FXIII deficiency cases, we found that two *HLA-DPB1* polymorphisms (rs1126504 and rs1126509) tended to decrease FXIII inhibitory titers as the number of variant alleles increased (Fig 2*A*, Tables 3 and S9). Similar results were obtained from the levels of anti-FXIII autoantibodies measured by ELISA in each genotype of 44 candidate alleles (Tables 4 and S9). On the other hand, among polymorphisms with significantly higher frequencies in autoimmune FXIII deficiency cases, we also found that five polymorphisms, *HLA-B* (rs1050723), *MICA* (rs1131897), *BTNL2* (rs2076530), *HLA-DQB1* (rs41544112), and *HLA-DPB1* (rs1042131), tended to increase levels of anti-FXIII-A autoantibodies measured using ICT as the number of variant alleles increased (Fig 2*B*, Tables 5 and S9).

## Discussion

To the best of our knowledge, this is the first study to identify the genetic factors associated with the development of anti-FXIII autoantibodies in autoimmune FXIII deficiency. In this

**Table 3. FXIII inhibitors in each genotypes of MHC class I and II molecules and their associated genes.**

| Frequency | Chr | Pos | Ref | Var | Gene ID | dbSNP_ID | N | | | FXIII inhibitor (BU) | | |
| --- | --- | --- | --- | --- | --- | --- | --- | --- | --- | --- | --- | --- |
| | | | | | | | | | | | Median | |
| | | | | | | | Ref | Hetero | Homo | Ref | Hetero | Homo |
| High | 2 | 204737478 | C | G | CTLA4 | rs74808460 | 19 | 1 | 0 | 6.90 | 27.00 | |
| High | 6 | 29912856 | A | T | HLA-A | rs2231119 | 1 | 3 | 16 | 6.90 | 0.80 | 9.10 |
| High | 6 | 31236853 | G | A | HLA-C | rs1065711 | 0 | 4 | 16 | | 3.60 | 12.30 |
| High | 6 | 31238155 | G | A | HLA-C | rs1050328 | 3 | 6 | 11 | 21.00 | 2.40 | 9.70 |
| High | 6 | 31323321 | G | A | HLA-B | rs1050723 | 15 | 5 | 0 | 6.40 | 9.70 | |
| Low | 6 | 31324506 | C | T | HLA-B | rs1050388 | 19 | 1 | 0 | 8.50 | 0.70 | |
| Low | 6 | 31324549 | T | C | HLA-B | rs1050570 | 17 | 2 | 1 | 0.75 | 5.80 | 51.50 |
| High | 6 | 31379134 | C | G | MICA | rs1131897 | 15 | 5 | 0 | 6.40 | 9.70 | |
| Low | 6 | 31379807 | C | T | MICA | rs1051798 | 19 | 1 | 0 | 8.50 | 0.90 | |
| Low | 6 | 31379823 | C | G | MICA | rs1051799 | 19 | 1 | 0 | 8.50 | 0.90 | |
| High | 6 | 32362741 | C | T | BTNL2 | rs28362677 | 10 | 8 | 2 | 8.30 | 7.45 | 25.75 |
| High | 6 | 32362745 | G | A | BTNL2 | rs28362678 | 10 | 8 | 2 | 8.30 | 7.45 | 25.75 |
| High | 6 | 32363816 | T | C | BTNL2 | rs2076530 | 4 | 11 | 5 | 3.25 | 8.50 | 9.70 |
| Low | 6 | 32549525 | C | G | HLA-DRB1 | rs111965977 | 18 | 2 | 0 | 6.65 | 27.35 | |
| Low | 6 | 32549531 | T | C | HLA-DRB1 | rs112796209 | 18 | 2 | 0 | 6.65 | 27.35 | |
| Low | 6 | 32629868 | A | G | HLA-DQB1 | rs1049088 | 16 | 4 | 0 | 9.10 | 6.10 | |
| High | 6 | 32629920 | C | T | HLA-DQB1 | rs41544112 | 15 | 5 | 0 | 6.40 | 9.70 | |
| Low | 6 | 32629936 | C | T | HLA-DQB1 | rs1049107 | 16 | 4 | 0 | 9.10 | 6.10 | |
| Low | 6 | 32629963 | C | T | HLA-DQB1 | rs1049100 | 16 | 4 | 0 | 9.10 | 6.10 | |
| Low | 6 | 32632745 | G | A | HLA-DQB1 | rs1063318 | 10 | 9 | 1 | 8.30 | 4.00 | 26.40 |
| High | 6 | 32632801 | G | A | HLA-DQB1 | rs3204373 | 12 | 7 | 1 | 8.30 | 4.00 | 26.40 |
| Low | 6 | 32632818 | T | G | HLA-DQB1 | rs1130368 | 19 | 1 | 0 | 6.90 | 51.50 | |
| Low | 6 | 32632832 | A | T | HLA-DQB1 | rs9274407 | 7 | 5 | 8 | 8.50 | 0.90 | 12.30 |
| Low | 6 | 33048457 | C | G | HLA-DPB1 | rs1126504 | 17 | 3 | 0 | 9.70 | 0.10 | |
| Low | 6 | 33048461 | T | A | HLA-DPB1 | rs1126509 | 17 | 3 | 0 | 9.70 | 0.10 | |
| High | 6 | 33048542 | C | T | HLA-DPB1 | rs1042121 | 1 | 1 | 18 | 6.90 | 9.70 | 7.45 |
| High | 6 | 33048602 | C | A | HLA-DPB1 | rs1042131 | 7 | 5 | 8 | 6.90 | 27.00 | 5.20 |
| Low | 6 | 33048661 | A | G | HLA-DPB1 | rs1042151 | 19 | 1 | 0 | 8.50 | 0.00 | |
| Low | 6 | 33048663 | G | A | HLA-DPB1 | rs1042153 | 19 | 1 | 0 | 8.50 | 0.00 | |

If variant allelic frequency (compared with that in the database) was significantly high, the term "High" was described in column 1. If the frequency was significantly low, the term "Low" was described in column 1. The missing values were displayed in a gray box.

study, we performed WES analysis of autoimmune FXIII deficiency and narrowed down the candidate alleles based on their allelic frequencies and the magnitude of damage caused by AA substitutions. We also investigated the relationship between the 44 selected candidate alleles and the FXIII inhibitory titers and/or the levels of anti-FXIII autoantibodies via ELISA and ICT. We found that two polymorphisms tended to decrease FXIII inhibitory titers as the number of variant alleles increased and these polymorphisms were significantly lower variant allele frequencies in autoimmune FXIII deficiency cases. On the other hand, we found that five polymorphisms tended to increase levels of anti-FXIII-A autoantibodies via ICT as the number of variant alleles increased and these polymorphisms were significantly higher frequencies in autoimmune FXIII deficiency cases. All these polymorphisms were exclusively found in the HLA class I and II molecules and their associated genes.

**Table 4. Anti-FXIII-A autoantibodies measured by ELISA in each genotypes of MHC class I and II molecules and their associated genes.**

| Frequency | Chr | Pos | Ref | Var | Gene ID | dbSNP_ID | N | | | Relative anti-FXIII-A autoantibodies against autoimmune FXIII deficiency-1 set to 1.0 (ELISA) | | |
|---|---|---|---|---|---|---|---|---|---|---|---|---|
| | | | | | | | | | | | Median | |
| | | | | | | | Ref | Hetero | Homo | Ref | Hetero | Homo |
| High | 2 | 204737478 | C | G | *CTLA4* | rs74808460 | 19 | 1 | 0 | 1.10 | 1.67 | |
| High | 6 | 29912856 | A | T | *HLA-A* | rs2231119 | 1 | 3 | 16 | 0.90 | 1.29 | 1.17 |
| High | 6 | 31236853 | G | A | *HLA-C* | rs1065711 | 0 | 4 | 16 | | 0.77 | 1.35 |
| High | 6 | 31238155 | G | A | *HLA-C* | rs1050328 | 3 | 6 | 11 | 1.56 | 0.70 | 1.24 |
| High | 6 | 31323321 | G | A | *HLA-B* | rs1050723 | 15 | 5 | 0 | 1.24 | 1.10 | |
| Low | 6 | 31324506 | C | T | *HLA-B* | rs1050388 | 19 | 1 | 0 | 1.24 | 0.30 | |
| Low | 6 | 31324549 | T | C | *HLA-B* | rs1050570 | 17 | 2 | 1 | 1.05 | 1.32 | 2.04 |
| High | 6 | 31379134 | C | G | *MICA* | rs1131897 | 15 | 5 | 0 | 1.24 | 1.10 | |
| Low | 6 | 31379807 | C | T | *MICA* | rs1051798 | 19 | 1 | 0 | 1.24 | 1.00 | |
| Low | 6 | 31379823 | C | G | *MICA* | rs1051799 | 19 | 1 | 0 | 1.24 | 1.00 | |
| High | 6 | 32362741 | C | T | *BTNL2* | rs28362677 | 10 | 8 | 2 | 1.25 | 1.17 | 1.04 |
| High | 6 | 32362745 | G | A | *BTNL2* | rs28362678 | 10 | 8 | 2 | 1.25 | 1.17 | 1.04 |
| High | 6 | 32363816 | T | C | *BTNL2* | rs2076530 | 4 | 11 | 5 | 0.92 | 1.24 | 1.05 |
| Low | 6 | 32549525 | C | G | *HLA-DRB1* | rs111965977 | 18 | 2 | 0 | 1.08 | 1.53 | |
| Low | 6 | 32549531 | T | C | *HLA-DRB1* | rs112796209 | 18 | 2 | 0 | 1.08 | 1.53 | |
| Low | 6 | 32629868 | A | G | *HLA-DQB1* | rs1049088 | 16 | 4 | 0 | 1.17 | 1.09 | |
| High | 6 | 32629920 | C | T | *HLA-DQB1* | rs41544112 | 15 | 5 | 0 | 1.24 | 1.05 | |
| Low | 6 | 32629936 | C | T | *HLA-DQB1* | rs1049107 | 16 | 4 | 0 | 1.17 | 1.09 | |
| Low | 6 | 32629963 | C | T | *HLA-DQB1* | rs1049100 | 16 | 4 | 0 | 1.17 | 1.09 | |
| Low | 6 | 32632745 | G | A | *HLA-DQB1* | rs1063318 | 10 | 9 | 1 | 1.15 | 1.10 | 1.87 |
| High | 6 | 32632801 | G | A | *HLA-DQB1* | rs3204373 | 12 | 7 | 1 | 1.15 | 1.10 | 1.87 |
| Low | 6 | 32632818 | T | G | *HLA-DQB1* | rs1130368 | 19 | 1 | 0 | 1.10 | 2.04 | |
| Low | 6 | 32632832 | A | T | *HLA-DQB1* | rs9274407 | 7 | 5 | 8 | 1.10 | 1.00 | 1.35 |
| Low | 6 | 33048457 | C | G | *HLA-DPB1* | rs1126504 | 17 | 3 | 0 | 1.29 | 0.04 | |
| Low | 6 | 33048461 | T | A | *HLA-DPB1* | rs1126509 | 17 | 3 | 0 | 1.29 | 0.04 | |
| High | 6 | 33048542 | C | T | *HLA-DPB1* | rs1042121 | 1 | 1 | 18 | 0.90 | 1.05 | 1.27 |
| High | 6 | 33048602 | C | A | *HLA-DPB1* | rs1042131 | 7 | 5 | 8 | 1.05 | 1.67 | 0.70 |
| Low | 6 | 33048661 | A | G | *HLA-DPB1* | rs1042151 | 19 | 1 | 0 | 1.24 | 0.04 | |
| Low | 6 | 33048663 | G | A | *HLA-DPB1* | rs1042153 | 19 | 1 | 0 | 1.24 | 0.04 | |

The production of autoantibodies is considered to be caused by the disruption of the mechanism of immune tolerance [21,22]. The specific disruption mechanism has not yet been completely elucidated; however, several mechanisms have been proposed, such as abnormal apoptosis, abnormal regulatory T-cells, and molecular homology with foreign antigens. We found four *F13A1* variants and one *F13B* variant exhibiting single AA changes. Among these variants, the allelic frequency of one variant (rs5982, p.Pro565Leu) was lower in autoimmune FXIII deficiency cases than that registered in the five databases, while that of one variant (rs76451285, p.Ala395Val) was higher. However, Ala395 was hidden inside the molecule, while Pro565 was exposed outside [23]. Of the 20 cases of autoimmune FXIII deficiency, 12 had reference alleles at the *F13A1* variant (rs5982), while 8 cases exhibited heterozygosity. If the variant was associated with the development of anti-FXIII autoantibodies, the levels of the anti-FXIII autoantibodies of heterozygotes would be lower than those of the reference allele

**Table 5. Anti-FXIII-A autoantibodies measured by ICT in each genotypes of MHC class I and II molecules and their associated genes.**

| Frequency | Chr | Pos | Ref | Var | Gene ID | dbSNP_ID | N | | | Anti-FXIII-A autoantibodies (ICT, AU) | | |
|---|---|---|---|---|---|---|---|---|---|---|---|---|
| | | | | | | | | | | Median | | |
| | | | | | | | Ref | Hetero | Homo | Ref | Hetero | Homo |
| High | 2 | 204737478 | C | G | CTLA4 | rs74808460 | 19 | 1 | 0 | 0.46 | 0.61 |  |
| High | 6 | 29912856 | A | T | HLA-A | rs2231119 | 1 | 3 | 16 | 0.35 | 0.45 | 0.50 |
| High | 6 | 31236853 | G | A | HLA-C | rs1065711 | 0 | 4 | 16 |  | 0.40 | 0.50 |
| High | 6 | 31238155 | G | A | HLA-C | rs1050328 | 3 | 6 | 11 | 0.61 | 0.40 | 0.46 |
| High | 6 | 31323321 | G | A | HLA-B | rs1050723 | 15 | 5 | 0 | 0.35 | 0.93 |  |
| Low | 6 | 31324506 | C | T | HLA-B | rs1050388 | 19 | 1 | 0 | 0.49 | 0.05 |  |
| Low | 6 | 31324549 | T | C | HLA-B | rs1050570 | 17 | 2 | 1 | 0.45 | 0.48 | 0.65 |
| High | 6 | 31379134 | C | G | MICA | rs1131897 | 15 | 5 | 0 | 0.35 | 0.93 |  |
| Low | 6 | 31379807 | C | T | MICA | rs1051798 | 19 | 1 | 0 | 0.49 | 0.23 |  |
| Low | 6 | 31379823 | C | G | MICA | rs1051799 | 19 | 1 | 0 | 0.49 | 0.23 |  |
| High | 6 | 32362741 | C | T | BTNL2 | rs28362677 | 10 | 8 | 2 | 0.41 | 0.53 | 0.35 |
| High | 6 | 32362745 | G | A | BTNL2 | rs28362678 | 10 | 8 | 2 | 0.41 | 0.53 | 0.35 |
| High | 6 | 32363816 | T | C | BTNL2 | rs2076530 | 4 | 11 | 5 | 0.14 | 0.49 | 0.64 |
| Low | 6 | 32549525 | C | G | HLA-DRB1 | rs111965977 | 18 | 2 | 0 | 0.48 | 0.50 |  |
| Low | 6 | 32549531 | T | C | HLA-DRB1 | rs112796209 | 18 | 2 | 0 | 0.48 | 0.50 |  |
| Low | 6 | 32629868 | A | G | HLA-DQB1 | rs1049088 | 16 | 4 | 0 | 0.47 | 0.51 |  |
| High | 6 | 32629920 | C | T | HLA-DQB1 | rs41544112 | 15 | 5 | 0 | 0.45 | 0.93 |  |
| Low | 6 | 32629936 | C | T | HLA-DQB1 | rs1049107 | 16 | 4 | 0 | 0.47 | 0.51 |  |
| Low | 6 | 32629963 | C | T | HLA-DQB1 | rs1049100 | 16 | 4 | 0 | 0.47 | 0.51 |  |
| Low | 6 | 32632745 | G | A | HLA-DQB1 | rs1063318 | 10 | 9 | 1 | 0.48 | 0.49 | 0.22 |
| High | 6 | 32632801 | G | A | HLA-DQB1 | rs3204373 | 12 | 7 | 1 | 0.48 | 0.49 | 0.22 |
| Low | 6 | 32632818 | T | G | HLA-DQB1 | rs1130368 | 19 | 1 | 0 | 0.46 | 0.65 |  |
| Low | 6 | 32632832 | A | T | HLA-DQB1 | rs9274407 | 7 | 5 | 8 | 0.35 | 0.45 | 0.53 |
| Low | 6 | 33048457 | C | G | HLA-DPB1 | rs1126504 | 17 | 3 | 0 | 0.49 | 0.04 |  |
| Low | 6 | 33048461 | T | A | HLA-DPB1 | rs1126509 | 17 | 3 | 0 | 0.49 | 0.04 |  |
| High | 6 | 33048542 | C | T | HLA-DPB1 | rs1042121 | 1 | 1 | 18 | 0.35 | 0.64 | 0.48 |
| High | 6 | 33048602 | C | A | HLA-DPB1 | rs1042131 | 7 | 5 | 8 | 0.35 | 0.61 | 0.53 |
| Low | 6 | 33048661 | A | G | HLA-DPB1 | rs1042151 | 19 | 1 | 0 | 0.49 | 0.04 |  |
| Low | 6 | 33048663 | G | A | HLA-DPB1 | rs1042153 | 19 | 1 | 0 | 0.49 | 0.04 |  |

because the allelic frequency of the variant in autoimmune FXIII deficiency was lower than that registered in the databases. However, in heterozygous cases, the FXIII inhibitory titers and anti-FXIII autoantibody levels were higher than those in the reference allele cases, and there were no significant differences between these two groups. Therefore, it is unlikely that the *F13A1* variants are involved in the production of autoantibodies; however, this need to be investigated further in future studies.

CTLA-4 is an important negative regulator of the immune system, exhibiting several polymorphisms associated with susceptibility to autoimmune diseases [22,24]. One of the polymorphisms (rs231775, p.Thr17Ala) was detected at a significantly higher frequency in all AHA cases [9,11]. In the present study, the allelic frequencies of this polymorphism in autoimmune FXIII deficiency cases was 0.58, which was comparable to that of the AHA cases. However, the frequency in Asia or East Asia was much higher than the global average and almost the same as that in all the autoimmune FXIII deficiency cases. On the other hand, we found other polymorphism (rs74808460) that may be associated with the development of autoantibodies in the

patients. Whether these polymorphisms are actually associated with the development of auto-antibodies should be confirmed by increasing the number of patients.

Various HLA alleles are known to contribute to susceptibility/protection to autoimmunity and play a definite role in the regulation of T-cell signaling [21,22]. Here, we identified two *HLA-DPB1* polymorphisms (rs1126504 and rs1126509) associated with both FXIII inhibitory titers and levels of anti-FXIII autoantibodies measured by ELISA. However, these two polymorphisms might be on the same haplotype because the genotype pattern was exactly the same for each autoimmune FXIII deficiency patient.

We also identified five polymorphisms associated with the levels of anti-FXIII autoantibodies measured by ICT. These were, one *HLA-B* polymorphism (rs1050723), one *MICA* polymorphism (rs1131897), one *BTNL2* polymorphism (rs2076530), one *HLA-DQB1* polymorphism (rs41544112), and one *HLA-DPB1* polymorphism (rs1042131). Of these, the *HLA-B* and *MICA* polymorphisms might be on the same haplotype as well as two *HLA-DPB1* polymorphisms.

There are two possible causes for the differences between the autoantibody levels measured using ELISA and ICT. First, in the case of measuring autoantibody levels using ICT, especially when they are extremely high, FXIII in the sample that binds to the autoantibody becomes saturated and insufficient, resulting in insufficient quantification. Second, as the sample concentration used in ICT is much higher than that used in ELISA, it is possible that low-affinity autoantibodies are also detected. The amount of autoantibodies with higher affinity is considered to be similar to the results of ELISA. The results of the FXIII inhibitory titer and levels of autoantibodies measured using ELISA were in good agreement. Therefore, we hypothesized that the *HLA-DPB1* polymorphism (rs1126504) is important for the development of autoantibodies in autoimmune FXIII deficiency. However, ICT is effective at diagnosing autoimmune FXIII deficiency with a specificity of 0.87 and sensitivity of 0.94 [20], and polymorphisms associated with the levels of autoantibodies measured using ICT cannot be excluded from the candidate alleles; therefore, the results of ICT were also considered.

*HLA-B* is an HLA class I molecule, and *MICA* is its associated gene whose name is an abbreviation for "major histocompatibility complex (MHC: synonymous with HLA) class I polypeptide-related sequence A." *HLA-DQB1* and *HLA-DPB1* are HLA class II molecules and *BTNL2*, the associated gene of HLA class II molecules is also known as "butyrophilin-like protein 2" [25]. BTNL2 shares sequence homology with the B7 proteins that regulate T-cell activation and tolerance [26,27]. The *BTNL2* mutation has been recently associated with inflammatory autoimmune diseases such as sarcoidosis and myositis [28–30]. In fact, the *BTNL2* polymorphism (rs2076530) has been registered in the dbSNP database as a risk factor for sarcoidosis.

The frequencies of *CTLA-4*, *HLA-DRB1*, and *HLA-DQB1* alleles in AHA cases have been previously compared to those of the healthy controls [9,11]. Here, we compared the frequencies of these genes in the autoimmune FXIII deficiency cases with the frequencies registered in the five databases. However, the frequencies of *CTLA-4* and *HLA-DRB1* were not significantly different between the autoimmune FXIII deficiency cases and those reported by the database. The frequencies of *HLA-DQB1* polymorphism (rs41544112), which is characteristic of *DQB1\*06*, were higher in the autoimmune FXIII deficiency cases than those reported in the database. Therefore, the genetic risk factors of autoimmune FXIII deficiency may differ from those of AHA, as the frequencies of *DQB1\*0502* have been reported to be higher than those of the healthy controls.

## Limitations

As autoimmune FXIII deficiency is a rare disease, it was difficult to recruit a large number of patients and only 20 cases were analyzed in this study. Furthermore, due to the involvement of

a rare polymorphism, it was difficult to obtain an appropriate number of patients to perform the statistical analysis. Therefore, it is necessary to verify the results of this study by conducting further investigations with larger sample sizes. In addition, HLA consists of several similar polymorphisms and pseudogenes, which cannot be easily distinguished from each other. In this study, we identified at least two cases in which two polymorphisms were thought to be on the same haplotype. Therefore, future studies can employ specific HLA-typing techniques to obtain novel results. Lastly, as the information available on this rare disease is very limited, the results obtained in this study may require further verification. However, these results may be used as a reference to elucidate the mechanism of pathogenesis of this disease in future studies.

## Conclusions

In this study, we found that *HLA-DPB1* polymorphisms were important for the development of autoantibodies in autoimmune FXIII deficiency, while the potential involvement of the *HLA-DQB1* and *BTNL2* polymorphisms was also indicated by the results. We believe that these genetic factors, along with other genetic factors and environmental factors, such as aging, together result in the development of autoimmune FXIII deficiency.

## Supporting information

**S1 Fig. Breakdown of identified variants.** *A*, Number of variants per chromosomes in each case. *B*, Number of heterozygous and homozygous variants in each case. *C*, Number of each genetic variant type in each case. *D*, Number of each codon mutation type in each case. (TIF)

**S2 Fig. Breakdown of candidate 3.** *A*, Number of variants per chromosomes in each case. *B*, Number of heterozygous and homozygous variants in each case. *C*, Number of each genetic variant type in each case. *D*, Number of each codon mutation type in each case. (TIF)

**S1 Table. Summary of FXIII tests in 20 autoimmune FXIII deficiency cases.** (XLSX)

**S2 Table. Comparison of the distribution of the number of autoimmune FXIII deficiency cases used in this study and the population of each prefecture in Japan.** Japanese population data was obtained from "Population by Sex and Sex ratio for Prefectures—Total population, Japanese population, October 1, 2016" in portal site of official statistics of Japan (https://www. e-stat.go.jp/en/stat-search/files?page=1&layout=datalist&toukei=00200524&tstat= 000000090001&cycle=7&year=20160&tclass1=000001011679&tclass2val=0) and modified the layout a little. Two-tailed Fisher's exact test was used to compare differences in distribution between the Japanese population and the autoimmune FXIII deficiency cases used in this study. (XLSX)

**S3 Table. Variants in *F13A1*, *F13B*, *CTLA4*, *HLA-DRB1*, and *HLA-DQB1* in autoimmune FXIII deficiency cases and its allelic frequency compared with that registered in five databases concerning total and (East) Asia.** When the OR of autoimmune FXIII deficiency against each (East) Asia database was > 1.5 or < 0.67, the OR was represented in bold letters. When the P-value was < 1.00E-8, the value was represented as "<1.00E-8" with a bold letter. Following three cases, chromosome number (Chr), position (Pos), reference nucleotide sequence (Ref), variant nucleotide sequence (Var), and gene ID (Gene ID) were represented in bold letters when the codon mutation type was single AA change. 1) In the case of the OR of

autoimmune FXIII deficiency in each database was > 1.5 or < 0.67 no defect was present in all databases. 2) In the case of the OR of autoimmune FXIII deficiency to the non-defective database was all > 1.5 or < 0.67 when there were some defects. 3) In the case of all databases were missing.
(XLSX)

**S4 Table. Number of codon mutations of genes associated with GO terms "T cell activation," "antigen presentation," or "immune tolerance" in each case.** Total number > 100 was represented by a bold letter.
(XLSX)

**S5 Table. Variants of genes associated with GO terms "T cell activation," "antigen presentation," or "immune tolerance" in autoimmune FXIII deficiency cases.** When the OR of autoimmune FXIII deficiency against each (East) Asia database was > 1.5 or < 0.67, the OR was represented in bold letters. When the P-value was < 1.00E-8, the value was represented as "<1.00E-8" with a bold letter. Following three cases, chromosome number (Chr), position (Pos), reference nucleotide sequence (Ref), variant nucleotide sequence (Var), and gene ID (Gene ID) were represented in bold letters when the codon mutation type was single AA change. 1) In the case of the OR of autoimmune FXIII deficiency to each database was > 1.5 or < 0.67 when there was no defect in all databases. 2) In the case of the OR of autoimmune FXIII deficiency to the non-defective database was > 1.5 or < 0.67 when there were some defects. 3) In the case of all databases were missing. Polymorphisms that do not have a second (Var2) or third variant (Var3) in autoimmune FXIII deficiency are displayed in a gray box.
(XLSX)

**S6 Table. Number of variant genes that thought to cause a damage in variants of S4 Table.** Total number of 20 or greater was represented by a bold letter.
(XLSX)

**S7 Table. Damaging mutations of genes associated with GO terms "T cell activation," "antigen presentation," or "immune tolerance" in autoimmune FXIII deficiency cases.** When the OR of autoimmune FXIII deficiency against each (East) Asia database was > 1.5 or < 0.67, the OR was represented in bold letters. When the P-value was < 1.00E-8, the value was represented as "<1.00E-8" with a bold letter. Following 3 cases, chromosome number (Chr), position (Pos), reference nucleotide sequence (Ref), variant nucleotide sequence (Var), and gene ID (Gene ID) were represented in bold letters when the codon mutation type was single AA change. 1) In the case of the OR of autoimmune FXIII deficiency to each database was > 1.5 or < 0.67 when there was no defect in all databases. 2) In the case of the OR of autoimmune FXIII deficiency to the non-defective database was > 1.5 or < 0.67 when there were some defects. 3) In the case of all databases were missing.
(XLSX)

**S8 Table. The list of the codon mutations that probably cause damage with variant allelic frequency < 0.01 in 20 autoimmune FXIII deficiency cases.** When the allelic frequency of the database was < 1.00E-2, the frequency was represented in bold letters. When the OR of autoimmune FXIII deficiency against each (East) Asia database was > 1.5 or < 0.67, the OR was represented in bold letters. When the P-value was < 1.00E-8, the value was represented as "<1.00E-8" with a bold letter. When the case numbers whose genotypes were "Homozygous" or "Heterozygous" were ≥ 10, chromosome number (Chr), position (Pos), reference nucleotide sequence (Ref), and variant nucleotide sequence (Var) were represented in bold letters.
(XLSX)

**S9 Table. FXIII inhibitors and anti-FXIII autoantibody levels measured by ELISA and ICT in each genotypes except for MHC class I and II molecules and their associated genes.** If variant allelic frequency compared with database was significantly high, the term "High" was described in column 1. If the frequency was significantly low, the term "Low" was described in column 1. The missing value are displayed in a gray box.
(XLSX)

## Acknowledgments

We would like to thank all members of the "Japanese Collaborative Research Group (JCRG) on Autoimmune Coagulation Factor Deficiencies." We would also like to thank Editage (www.editage.com) for English language editing.

## Author Contributions

**Conceptualization:** Tsukasa Osaki, Akitada Ichinose.

**Data curation:** Tsukasa Osaki, Masayoshi Souri, Akitada Ichinose.

**Formal analysis:** Tsukasa Osaki.

**Funding acquisition:** Akitada Ichinose.

**Investigation:** Tsukasa Osaki, Masayoshi Souri, Akitada Ichinose.

**Methodology:** Tsukasa Osaki.

**Project administration:** Akitada Ichinose.

**Supervision:** Akitada Ichinose.

**Visualization:** Tsukasa Osaki.

**Writing – original draft:** Tsukasa Osaki.

**Writing – review & editing:** Tsukasa Osaki, Masayoshi Souri, Akitada Ichinose.

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
