## [Decision Letter · Decision Letter 0]

10 May 2021

PONE-D-21-03903

Whole-exome sequencing analysis of autoimmune coagulation factor XIII/13 deficiencies reveals the importance of human leucocyte antigen class I and II genes and their associated genes

PLOS ONE

Dear Dr. Osaki,

Thank you for submitting your manuscript to PLOS ONE. After careful consideration, we feel that it has merit but does not fully meet PLOS ONE’s publication criteria as it currently stands. Therefore, we invite you to submit a revised version of the manuscript that addresses the points raised during the review process.

As you can see from the comments of the reviewers, a significant revision of the manuscript is expected for this article to proceed to the next round of evaluation/ selection.

We look forward to receiving your revised manuscript.

Kind regards,

Arijit Biswas

Academic Editor

PLOS ONE

Additional Editor Comments:

The paper by Osaki et al addressing autoimmune FXIII deficiencies has now been reviewed by two reviewers. Both reviewers have been highly critical of the article. While one reviewer has outright rejected the article, the other reviewer has left open the scope of a major revision. I agree with almost all the criticism raised by the reviewer. The article is inconsistent and lacking in several directions. The use of incorrect nomenclature can still be rectified and the grammar improved. However, the article suffers from poor execution with respect to statistical evaluation, explaining patient selection, clinical profile, genotypic selection etc. However, in spite of the deficiencies, owing to the data that is of interest to the FXIII specific audience I am willing to give the authors a chance to address the issues raised by the reviewers. Only and only if the authors satisfactorily address these issues will this article being continued to the next round of evaluation/ selection.

Journal Requirements:

2. In your Methods section, please provide additional information about the participant recruitment method and the demographic details of your participants. Please ensure you have provided sufficient details to replicate the analyses such as:

a) the institution(s) participants were recruited from,

b) the recruitment date range (month and year),

c) a description of any inclusion/exclusion criteria that were applied to participant recruitment,

d) a table of relevant demographic details,

e) a statement as to whether your sample can be considered representative of a larger population

3. Please provide catalog numbers, sources, and dilutions of anti-F13-A monoclonal antibody and the recombinant F13-A protein used in this study.

4. Please provide citations for PROVEAN and PolyPhen-2.

Reviewers' comments:

Reviewer's Responses to Questions

**Comments to the Author**

1. Is the manuscript technically sound, and do the data support the conclusions?

Reviewer #1: No

Reviewer #2: Partly

2. Has the statistical analysis been performed appropriately and rigorously? 

Reviewer #1: No

Reviewer #2: No

3. Have the authors made all data underlying the findings in their manuscript fully available?

Reviewer #1: Yes

Reviewer #2: Yes

4. Is the manuscript presented in an intelligible fashion and written in standard English?

Reviewer #1: No

Reviewer #2: Yes

5. Review Comments to the Author

Reviewer #1: The article presents a study to deep in the genetic basis of the Acquired FXIII deficiency. The authors present the results of a cohort of 20 patients. They sequence the whole exome, and analyse different candidate variants to see if there is a correlation with the FXIII inhibitory titers, and anti-FXIII autoantibodies.

The first thing that surprise is the nomenclature used to refer to the disease. The authors, in a previous article “Recommendation for ISTH/SSC Criterion 2015 for autoimmune acquired factor XIII/13 deficiency” (Thromb Haemost 2016; 116:772-774) refer to this deficiency as AAXIII/13D instead of AiF13D. The same article explains that Coagulation factor XIII is abbreviated as FXIII, instead of F13. Having said that, and as a general evaluation, the Introduction is weak and little argued. In page 4, authors state that “AiF13D occurs as a result of the spontaneous production of autoantibodies against endogenous F13”, while in their previous article (Thromb Haemost 2016; 116:772-774) stated that “About half of AAXIII/13D cases are idiopathic, while the remaining half have an underlying disease(s)” and that there are different pathological mechanisms. In page 5, there are presented part of the results, that not have to be in the introduction.

The Material and Methods section is presented in an inaccurate manner. There is no information about the origin of the samples nor the inclusion criteria. It is not stated if they have to present low FXIII levels and no other alteration in the coagulation factors, or if they have to present anti-FXIII autoantibodies. Only “Patients with severe bleeding who did not have a personal or family history of bleeding were recruited for this study” is stated in page 6. The information about the NGS library and template preparation is insufficient, as AmpliSeq Library Kit is not a specific protocol for WES. In page 8, in the Allele call thresholds section, it is specified that “SNPs with allele frequencies of 95% or higher were considered homozygous for that allele, and those with allele frequencies of 30% to 70% were heterozygous. Allele frequencies of less than 30% and between 70% and 95% were ignored.” Those SNPs with allele frequencies between 70% and 95% are probably heterozygous or homozygous, and it is necessary to justify why they were ignored.

The Statistical analysis section is not enough, as most of the results presented are based on this type of analysis. The statistical approach described is incomplete and, based on the Results section and the information displayed on the tables, possibly incorrect.

-No clear information is given about the strategy followed to compare groups, such as which groups are being compared, or which variables are being considered. This information has to be deduced by the reader from other sections.

-The use of the Kruskal-Wallis non-parametric test is not argued. Moreover, this test is used to compare two or more independent samples and, again, based on the results presented in the tables, only comparisons of two groups are being shown. Therefore, the use of the Kruskal-Wallis test would not be suitable in these cases. Alternatively, and with the aim to conduct a basic statistical analysis to compare two groups, the Mann-Whitney non-parametric test or the parametric T-test could be applied.

-The ODDS ratio calculation is not described and the selection of the relevant thresholds (1.5 and 0.67) is not explicitly justified. In addition, empty cells in tables is not recommended: ODDS ratios out of the considered range (1.5 and 0.67) should be shown if they have been calculated, either in a main table or a supplementary.

-It is necessary to clarify the frequency of what is exactly being compared between the sample of acquired FXIII deficiency and the general population cohort. Generally, the available information from population panels is given as allelic or genotypic frequencies. For this analysis, in the Materials and Methods section, it is not described whether allelic or genotypic frequencies are calculated for the variants identified in the acquired FXIII deficiency sample and it is not specified which one of the latter is being compared. In the case that the raw number of variants detected in the acquired FXIII deficiency sample is being used, then the comparison is not adequate, as variants in this group may be either homozygous (two alternative alleles) or heterozygous (one alternative allele), which is not the frequency of alleles nor genotypes.

- In page 20, the meaning of the sentence “We selected variants whose numbers of the top two genotypes were three or more so that they could be compared by statistical processing” should be clarified and argued. What does “numbers” mean in this sentence? Number of variants? Of alleles? Of carriers (homozygous and heterozygous)? The “statistical processing” refers to the statistical power of this analysis? Then, it should be justified the suitability of “three numbers” to achieve certain statistical power and which the latter is.

About HLA, class I and II are not a gene (as stated in the abstract), it is a group of genes which polymorphisms determine the haplotype. As it is a very polymorphic locus, and there are different highly homologous pseudogenes, the alignment of the sequences of this area is very complex. For this reason, WES is not a good approximation, and it is necessary to use specific sequencing techniques and analysis software to establish the HLA typing.

Finally, the English have to be revised, as there are sentences that result difficult to understand. The quality of the figures is low and they are difficult to read.

For all the stated in this evaluation, I consider that the article is not suitable for publication in PlosOne.

Reviewer #2: The paper by Osaki et al reports the whole-exome sequencing analysis of autoimmune coagulation factor XIII (FXIII) deficiencies and reveals the association of certain leukocyte antigen class I and II genes and their associated genes with the development of autoantibodies.

The report is of potential interest, however, major corrections must be made before potential publication of this manuscript.

1/ The nomenclature of FXIII/FXIII deficiency used in the paper is unacceptable. FXIII, a coagulation factor (protein), cannot be referred to as „F13” -that is the nomenclature of the FXIII gene. Moreover, autoimmune FXIII deficiencies cannot be referred to as „AiF13D”. Also, deficiency subtypes used by the authors, e.g. Aa, Ab and B are not widely accepted. The authors must use the nomenclature that has been provided by the Scientific and Standardization Committee of the ISTH (Kohler et al, JTH 2011;9:1404-6). Autoimmune FXIII deficiency should be described according to a recent review paper published in JTH (Muszbek et al, JTH 2018; 16:822-32).

2/ Patient population is not well described. The paper must clearly describe how the patients were selected, list their symptoms, selection criteria, as well as main laboratory findings including FXIII activity and FXIII antigen levels (preferably FXIII-A2B2 and FXIII-B). Moreover, although a fraction of patients might have idiopathic autoimmune FXIII deficiency, one would expect to see association with cancer or autoimmunity, pregnancy, etc in others. Relevant information related to this must be provided.

3/ I believe that the associations described in the paper between certain tested genotypes and autoimmune FXIII deficiency were not proven to be causal in this paper,thus, the importance of the findings must remain limited in the Discussion and in the title of the manuscript.

4/ Associations described on page 20 (see „Association of selected candidates with FXIII inhibitory titers and/or anti-FXIII-A autoantibody levels” as well as Figure 2) are grossly underpowered statistically. In order to perform such analysis, statistically, at least 20-30 individuals per group must be present, otherwise conclusions may be misleading. In fact, as it is clear from Figure 2, results in all groups show an overlap, but due to the very low number of samples in some groups, results of the calculations are misleading. This section of the manuscript must be omitted or the number of patients/ group must be increased in order to provide correct statistical calculations.

5/ The manuscript should be more concise and I believe it could be significantly shortened to enhance clarity.

6. PLOS authors have the option to publish the peer review history of their article (what does this mean?). If published, this will include your full peer review and any attached files.

Reviewer #1: No

Reviewer #2: No

---

## [Author Response · Author response to Decision Letter 0]

24 Jun 2021

June 24, 2021

Dear Dr. Arijit Biswas:

We appreciate reviewer’s comments very much, for they help us to improve the quality of our paper considerably. According to the reviewer’s comments, the manuscript has been revised with yellow backgrounds and red letters.

Before responding point-by-point, we would like to draw your attention to two major additional changes that were revised according to the reviewers’ point out. First, we changed the title to enhance its readability from "Whole-exome sequencing analysis of autoimmune coagulation factor XIII/13 deficiencies reveals the importance of human leucocyte antigen class I and II genes and their associated genes" to " Important roles of the human leukocyte antigen class I and II molecules and their associated genes in the autoimmune coagulation factor XIII/13 deficiency via whole-exome sequencing analysis". Second, we added “Limitation” in the “Discussion” section to prevent readers from misunderstanding.

Response to Journal Requirements

Comment 1: Please ensure that your manuscript meets PLOS ONE's style requirements, including those for file naming. The PLOS ONE style templates can be found at https://journals.plos.org/plosone/s/file?id=wjVg/PLOSOne_formatting_sample_main_body.pdf and https://journals.plos.org/plosone/s/file?id=ba62/PLOSOne_formatting_sample_title_authors_affiliations.pdf

Reply to Comment 1: We followed the rules and worked carefully to ensure a style suitable for publication.

Comment 2: In your Methods section, please provide additional information about the participant recruitment method and the demographic details of your participants. Please ensure you have provided sufficient details to replicate the analyses such as:

Comment 2-1: a) the institution(s) participants were recruited from

Reply to Comment 2-1: According to Editor’s comment, we mentioned the institution in the “Results” section under the heading “WES analysis of AiF13D” (p.14, lane 9), but could not describe details without the consent of the patient. However, of the 20 patients, only 2 patients from the same institution were relatively unbiased.

Comment 2-2: b) the recruitment date range (month and year)

Reply to Comment 2-2: According to Editor’s comment, we described the recruitment date range (month and year) in the “Materials and methods” section under the heading “Clinical samples” (p.7, lane 4).

Comment 2-3: c) a description of any inclusion/exclusion criteria that were applied to participant recruitment

Reply to Comment 2-3: According to Editor’s comment, we described the inclusion/exclusion criteria in “Materials and methods” section under headings “Clinical samples” (p.7, lanes 2–7) and “NGS library and template preparation” (p.8, lanes 1–10).

Comment 2-4: d) a table of relevant demographic details

Reply to Comment 2-4: According to Editor’s comment, we obtained the table of relevant demographic details from "Population by Sex and Sex ratio for Prefectures - Total population, Japanese population, October 1, 2016" in portal site of official statistics of Japan (https://www.e-stat.go.jp/en/stat-search/files?page=1&layout=datalist&toukei=00200524&tstat=000000090001&cycle=7&year=20160&tclass1=000001011679&tclass2val=0) (p.7, lanes 11–15) and added S2 Table.

Comment 2-5: e) a statement as to whether your sample can be considered representative of a larger population

Reply to Comment 2-5: Following the Editor’s comment, we mentioned whether our sample can be considered representative of a larger population in the “Results” section under the heading “WES analysis of AiF13D” (p.14, lanes 9–11). As stated in the manuscript, the distribution of AiF13D cases by region is not significantly different from the distribution of Japanese, so we considered that the sample was representative of a larger population.

Comment 3-1: Please provide catalog numbers, sources, and dilutions of anti-F13-A monoclonal antibody used in this study.

Reply to Comment 3-1: We could not provide catalog number of the anti-FXIII/13-A monoclonal antibody because it was not a commercial available. However, we described the source of the antibody in the “Materials and methods” section under the heading “Materials” (p.6, lanes 11–12). We also described the dilutions of the antibody in the “Results” section under the heading “Detection of anti-FXIII/13-A autoantibodies using enzyme-linked immunosorbent assay (ELISA)” (p.12, lane 3).

Comment 3-2: Please provide catalog numbers, sources, and dilutions of the recombinant F13-A protein used in this study.

Reply to Comment 3-2: We could not provide catalog number of the recombinant FXIII/13-A because it was not a commercial available. However, we described the source of the recombinant protein in the “Materials and methods” section under the heading “Materials” (p.6, lane 10). We had described the dilutions of the recombinant protein in the “Results” section under the heading “Detection of anti-FXIII/13-A autoantibodies using enzyme-linked immunosorbent assay (ELISA)” in the original manuscript (p.12, lanes 1–2).

Comment 4: Please provide citations for PROVEAN and PolyPhen-2.

Reply to Comment 4: According to Editor’s comment, we described both citations for PROVEAN and PolyPhen-2 in the “Materials and methods” section under the heading “Ion Torrent data analysis” (p.10, lanes 11–12).

Comment 5: We note that the grant information you provided in the ‘Funding Information’ and ‘Financial Disclosure’ sections do not match. When you resubmit, please ensure that you provide the correct grant numbers for the awards you received for your study in the ‘Funding Information’ section.

Reply to Comment 5: We have confirmed that we are providing the correct grant number for the award we received for our study in the ‘Funding Information’ section.

Response to Reviewer #1

Comment 1-1: The first thing that surprise is the nomenclature used to refer to the disease. The authors, in a previous article “Recommendation for ISTH/SSC Criterion 2015 for autoimmune acquired factor XIII/13 deficiency” (Thromb Haemost 2016; 116:772–774) refer to this deficiency as AAXIII/13D instead of AiF13D.

Reply to Comment 1-1: As pointed out by Reviewer # 1, abbreviations different from the previous article might confuse the reader, but since the disease name designated as an intractable disease by the Japanese Ministry of Health, Labour and Welfare was AiF13D, we used AiF13D as the abbreviation. We explained in the "Introduction" section of this manuscript (p.4, lanes 5–10).

Comment 1-2: The same article explains that Coagulation factor XIII is abbreviated as FXIII, instead of F13.

Reply to Comment 1-2: As Reviewer #1 pointed out, we abbreviated coagulation factor XIII as FXIII/13 instead of F13. The reason why we did not use FXIII is to avoid confusion with FVIII and FXII for medical safety measures, although we wrote in the manuscript (p.3, lanes 12–13).

Comment 1-3: Having said that, and as a general evaluation, the Introduction is weak and little argued.

Reply to Comment 1-3: As Reviewer # 1 pointed out, “Introduction” was weak, so we clarified the purpose of this study (p.6, lanes 3–6).

Comment 2: In page 4, authors state that “AiF13D occurs as a result of the spontaneous production of autoantibodies against endogenous F13”, while in their previous article (Thromb Haemost 2016; 116:772–774) stated that “About half of AAXIII/13D cases are idiopathic, while the remaining half have an underlying disease(s)” and that there are different pathological mechanisms.

Reply to Comment 2: In fact, as Reviewer # 1 says, about half of AiF13D cases were idiopathic and the other half had underlying disease (Table 1), but we did not know how underlying disease affects antibody production. Therefore, we assumed that AiF13D occurs as a result of the production of autoantibodies against endogenous FXIII/13, and a supplementary explanation was added to "Introduction" (p.4, lanes 12–15).

Comment 3: In page 5, there are presented part of the results that not have to be in the introduction.

Reply to Comment 3: Following the Reviewer #1’s suggestions, we have removed some of the results from the last paragraph of “Introduction” section and replaced with the purpose of this study (p.6, lanes 3–6).

Comment 4-1: The Material and Methods section is presented in an inaccurate manner. There is no information about the origin of the samples nor the inclusion criteria.

Reply to Comment 4-1: As Reviewer #1 pointed out, information about the origin of the samples and the inclusion criteria were inadequate, so we added them. We already listed the sex and age of the patient in Table 1 of original manuscript, but newly added the patient’s medical institution (prefecture) to S1 Table. We also described the inclusion criteria in “Materials and methods” section under headings “Clinical samples” (p.7, lanes 2–7) and “NGS library and template preparation” (p.8, lanes 1–10).

Comment 4-2: It is not stated if they have to present low FXIII levels and no other alteration in the coagulation factors, or if they have to present anti-FXIII autoantibodies. Only “Patients with severe bleeding who did not have a personal or family history of bleeding were recruited for this study” is stated in page 6.

Reply to Comment 4-2: Following the Reviewer #1’s recommendation, we added the patient's underlying disease and its FXIII/13 activity to Table 1, and also added the FXIII/13-A, F-XIII/13-B, and FXIII/13-A2B2 antigen level, FXIII/13 inhibitory titer, and anti-FXIII/13 autoantibody level to S1 Table.

Comment 5: The information about the NGS library and template preparation is insufficient, as AmpliSeq Library Kit is not a specific protocol for WES.

Reply to Comment 5: As Reviewer #1 pointed out, we explained the NGS library and template preparation in detail (p.8, lanes 10–16).

Comment 6: In page 8, in the Allele call thresholds section, it is specified that “SNPs with allele frequencies of 95% or higher were considered homozygous for that allele, and those with allele frequencies of 30% to 70% were heterozygous. Allele frequencies of less than 30% and between 70% and 95% were ignored.” Those SNPs with allele frequencies between 70% and 95% are probably heterozygous or homozygous, and it is necessary to justify why they were ignored.

Reply to Comment 6: As Reviewer #1 points out, between 70% and 95% are probably heterozygous or homozygous. Following a previous report (Daniel R et al. Forensic Sci Int Genet. 2015;14:50–60), we changed 10% to 90% to be heterozygous, 90% or greater to be homozygous, and less than 10% to be ignored (p.10, lane 16–p.11, lane 2). According to this criterion, the number of variants changed a little, so I changed it in Table 1, S4, S6, and S8 Tables. The relevant parts of the manuscript have been altered accordingly.

Comment 7: The Statistical analysis section is not enough, as most of the results presented are based on this type of analysis. The statistical approach described is incomplete and, based on the Results section and the information displayed on the tables, possibly incorrect. No clear information is given about the strategy followed to compare groups, such as which groups are being compared, or which variables are being considered. This information has to be deduced by the reader from other sections.

Reply to Comment 7: As Reviewer #1 pointed out, there was not enough information. Therefore, we added a detailed explanation to the subsection entitled “Statistical analysis” in the “Materials and methods” section (p.13, lane 16–p.14, lane 4). AiF13D case distribution comparison and relative risk allele frequency comparisons were performed using a chi-square test or two-tailed Fisher's exact test. Comparisons of FXIII/13 inhibitory titer or levels of anti-FXIII/13 autoantibodies between two allele groups were performed using the Mann-Whitney U-test.

Comment 8: The use of the Kruskal-Wallis non-parametric test is not argued. Moreover, this test is used to compare two or more independent samples and, again, based on the results presented in the tables, only comparisons of two groups are being shown. Therefore, the use of the Kruskal-Wallis test would not be suitable in these cases. Alternatively, and with the aim to conduct a basic statistical analysis to compare two groups, the Mann-Whitney non-parametric test or the parametric T-test could be applied.

Reply to Comment 8: As Reviewer #1 pointed out, we changed "Kruskal-Wallis test" to "Mann-Whitney U-test" (p.14, lanes 1–3).

Comment 9-1: The ODDS ratio calculation is not described and the selection of the relevant thresholds (1.5 and 0.67) is not explicitly justified.

Reply to Comment 9: According to the opinion of Reviewer #1, we described the odds ratio calculation (p.13, lanes 12–15) and thresholds (p.13, lanes 15–16) in “Statistical analysis” in “Materials and methods” section.

Comment 9-2: In addition, empty cells in tables is not recommended: ODDS ratios out of the considered range (1.5 and 0.67) should be shown if they have been calculated, either in a main table or a supplementary.

Reply to Comment 9-2: The empty cells in Table 2 and S3, S5, S7, and S8 Tables does not mean out of the considered range (1.5 and 0.67), and we could not calculate the odds ratio because variant frequency of the SNP was not registered in the database or the variant allele frequency registered in the database was 0 or the frequency of AiF13D was 1. In the tables, we added NA instead of the empty cells.

Comment 10-1: It is necessary to clarify the frequency of what is exactly being compared between the sample of acquired FXIII deficiency and the general population cohort. 

Reply to Comment 10-1: Follow the advice of Reviewer #1, we clarified that it is a comparison with the allelic frequencies in “Statistical analysis” in “Materials and methods” section (p.13, lane 16–p.14, lane 1).

Comment 10-2: Generally, the available information from population panels is given as allelic or genotypic frequencies. For this analysis, in the Materials and Methods section, it is not described whether allelic or genotypic frequencies are calculated for the variants identified in the acquired FXIII deficiency sample and it is not specified which one of the latter is being compared.

Reply to Comment 10-2: As Reviewer #1 stated, we did not clarify either the allelic frequencies or the genotypic frequencies, so we clarified it as the allelic frequencies (p.13, lane 16–p.14, lane 1).

Comment 10-3: In the case that the raw number of variants detected in the acquired FXIII deficiency sample is being used, then the comparison is not adequate, as variants in this group may be either homozygous (two alternative alleles) or heterozygous (one alternative allele), which is not the frequency of alleles nor genotypes.

Reply to Comment 10-3: We calculated the allele frequency of the variant as (2n+m)/40 (20; total patient number x 2; number of alleles per person, except chromosomes X and Y), where n is the number of patients with variant homozygotes and m is the number of patients with heterozygotes. We added the calculation method for variant allelic frequencies to the subsection entitled “Statistical analysis” in the “Materials and methods” section (p.13, lanes 5–12).

Comment 11-1: In page 20, the meaning of the sentence “We selected variants whose numbers of the top two genotypes were three or more so that they could be compared by statistical processing” should be clarified and argued. What does “numbers” mean in this sentence? Number of variants? Of alleles? Of carriers (homozygous and heterozygous)? 

Reply to Comment 11-1: We meant the number of carriers in each carrier group (heterozygote, and variant and reference allele homozygotes). However, it is difficult to understand, so we changed the expression (p.23, lanes 9–12).

Comment 11-2: The “statistical processing” refers to the statistical power of this analysis?

Reply to Comment 11-2: Yes, we meant the “statistical processing” refers to the statistical power of this analysis.

Comment 11-3: Then, it should be justified the suitability of “three numbers” to achieve certain statistical power and which the latter is.

Reply to Comment 11-3: As Reviewer # 1 mentions, there is no justification for the “three numbers”, so we analyzed all applicable candidates (p.23, lanes 12–14).

Comment 12-1: About HLA, class I and II are not a gene (as stated in the abstract), it is a group of genes which polymorphisms determine the haplotype.

Reply to Comment 12-1: As Reviewer #1 pointed out, HLA class I and II are not genes, so we corrected the notation (Title, p.3, lane 1, Title of Tables 3–5, p.29, lane 11, p.33, lanes 1–5, and Title of S9 Table).

Comment 12-2: As it is a very polymorphic locus, and there are different highly homologous pseudogenes, the alignment of the sequences of this area is very complex. For this reason, WES is not a good approximation, and it is necessary to use specific sequencing techniques and analysis software to establish the HLA typing.

Reply to Comment 12-2: We are grateful to Reviewer #1 for pointing out. We described the difficulty of aligning the HLA region with WES in “Limitation” with reference to the pointed out content (p.34, lanes 9–13).

Comment 13-1: Finally, the English have to be revised, as there are sentences that result difficult to understand.

Reply to Comment 13-1: As shown in the attachment (Certificate_of_editing-AKNOS_29.pdf), we have already asked Editage (www.editage.com) to edit the original manuscript to make it easier to understand in English. I requested that the revised points be edited separately.

Comment 13-2: The quality of the figures is low and they are difficult to read.

Reply to Comment 13-2: As Reviewer #1 pointed out, we increased the resolution of the figures.

Response to Reviewer #2

Comment 1: The nomenclature of FXIII/FXIII deficiency used in the paper is unacceptable.

Comment 1-1: FXIII, a coagulation factor (protein), cannot be referred to as “F13” that is the nomenclature of the FXIII gene.

Reply to Comment 1-1: As Reviewer #2 pointed out, we abbreviated coagulation factor XIII as FXIII/13 instead of F13. The reason why we did not use FXIII is to avoid confusion with FVIII and FXII for medical safety measures, although we wrote in the manuscript (p.3, lanes 12–13).

Comment 1-2: Moreover, autoimmune FXIII deficiencies cannot be referred to as “AiF13D”.

Reply to Comment 1-2: As Reviewer # 2 pointed out, autoimmune FXIII deficiencies may not be described as "AiF13D", but since the disease name designated as an intractable disease by the Japanese Ministry of Health, Labor and Welfare was AiF13D, we used AiF13D as the abbreviation. We explained in the "Introduction" section of this manuscript (p.4, lanes 5–10).

Comment 1-3: Also, deficiency subtypes used by the authors, e.g. Aa, Ab and B are not widely accepted. The authors must use the nomenclature that has been provided by the Scientific and Standardization Committee of the ISTH (Kohler et al, JTH 2011;9:1404–6). Autoimmune FXIII deficiency should be described according to a recent review paper published in JTH (Muszbek et al, JTH 2018; 16:822–32).

Reply to Comment 1-3: Following the instructions of Reviewer #2, we removed the description of Aa, Ab and B.

Comment 2-1: Patient population is not well described. The paper must clearly describe how the patients were selected, list their symptoms, selection criteria, as well as main laboratory findings including FXIII activity and FXIII antigen levels (preferably FXIII-A2B2 and FXIII-B).

Reply to Comment 2-1: As Reviewer #2 stated, patient information is important in interpreting the results. We described the selection criteria in “Materials and methods” section under headings “Clinical samples” (p.7, lanes 2–7) and “NGS library and template preparation” (p.8, lanes 1–10). We added the FXIII/13 activity to Table 1, and listed FXIII/13 antigen levels, including FXIII/13-A, FXIII/13-B, and FXIII/13-A2B2, in S1 Table.

Comment 2-2: Moreover, although a fraction of patients might have idiopathic autoimmune FXIII deficiency, one would expect to see association with cancer or autoimmunity, pregnancy, etc in others. Relevant information related to this must be provided.

Reply to Comment 2-2: As Reviewer #2 pointed out, information on the underlying disease is important. We added the underlying disease to Table 1.

Comment 3: I believe that the associations described in the paper between certain tested genotypes and autoimmune FXIII deficiency were not proven to be causal in this paper, thus, the importance of the findings must remain limited in the Discussion and in the title of the manuscript.

Reply to Comment 3: Following the instructions of Reviewer #2, we changed the title from "Whole-exome sequencing analysis of autoimmune coagulation factor XIII/13 deficiencies reveals the importance of human leucocyte antigen class I and II genes and their associated genes" to " Important roles of the human leukocyte antigen class I and II molecules and their associated genes in the autoimmune coagulation factor XIII/13 deficiency via whole-exome sequencing analysis". We also added “Limitations” in “Discussion” section (p.34, lanes 5–16).

Comment 4-1: Associations described on page 20 (see “Association of selected candidates with FXIII inhibitory titers and/or anti-FXIII-A autoantibody levels” as well as Figure 2) are grossly underpowered statistically. In order to perform such analysis, statistically, at least 20–30 individuals per group must be present, otherwise conclusions may be misleading. 

Reply to Comment 4-1: It is understandable that a total of 20 cases is not enough to perform a comparative analysis. However, as added to "Limitations" (p.34, lanes 5–9), AiF13D cases are rare diseases, so it is difficult to secure specimens. In particular, the polymorphism shown here has a polymorphism frequency of 0.26-folds lower than that of healthy subjects, so the number has decreased to three cases. However, we found it worthwhile to share information that at least three patients with alleles, which were more common in healthy subjects than patients, had lower inhibitory titers.

Comment 4-2: In fact, as it is clear from Figure 2, results in all groups show an overlap, but due to the very low number of samples in some groups, results of the calculations are misleading.

Reply to Comment 4-2: As Reviewer # 2 pointed out, Figure 2B is the same polymorphism as Figure 2A, so we omitted Figure 2B.

Comment 4-3: This section of the manuscript must be omitted or the number of patients/ group must be increased in order to provide correct statistical calculations.

Reply to Comment 4-3: As we wrote in the “Reply to Comment 4-1”, we left this section for informational purposes, although it was limited in number.

Comment 5: The manuscript should be more concise and I believe it could be significantly shortened to enhance clarity.

Reply to Comment 5: We eliminated duplication. For example, we deleted the last paragraph of “Introduction” section due to duplicate results. We have already asked Editage (www.editage.com) to edit the original manuscript to make it more concise and clearer, as shown in the attachment (Certificate_of_editing-AKNOS_29.pdf). We requested that the revised points be edited separately.

---

## [Decision Letter · Decision Letter 1]

17 Aug 2021

PONE-D-21-03903R1

Important roles of the human leukocyte antigen class I and II molecules and their associated genes in the autoimmune coagulation factor XIII/13 deficiency via whole-exome sequencing analysis

PLOS ONE

Dear Dr. Osaki,

Thank you for submitting your manuscript to PLOS ONE. After careful consideration, and taking into account both the points made by both the current reviewers, I feel that the changes made in the first round of revision are quite unsatisfactory (especially with respect to the nomenclature /statistical analysis aspects raised by Reviewer2 who in fact has suggested rejection of the manuscript). However, I would like to give the authors one last chance to improve upon the manuscript especially in the lines of the points raised by Reviewer2 after which I will make a final decision if to accept the article or not. Therefore, I invite you to submit a revised version of the manuscript that addresses the points raised during the review process.

We look forward to receiving your revised manuscript.

Kind regards,

Arijit Biswas

Academic Editor

PLOS ONE

Reviewers' comments:

Reviewer's Responses to Questions

**Comments to the Author**

1. If the authors have adequately addressed your comments raised in a previous round of review and you feel that this manuscript is now acceptable for publication, you may indicate that here to bypass the “Comments to the Author” section, enter your conflict of interest statement in the “Confidential to Editor” section, and submit your "Accept" recommendation.

Reviewer #2: (No Response)

Reviewer #3: (No Response)

2. Is the manuscript technically sound, and do the data support the conclusions?

Reviewer #2: No

Reviewer #3: Yes

3. Has the statistical analysis been performed appropriately and rigorously? 

Reviewer #2: No

Reviewer #3: Yes

4. Have the authors made all data underlying the findings in their manuscript fully available?

Reviewer #2: Yes

Reviewer #3: Yes

5. Is the manuscript presented in an intelligible fashion and written in standard English?

Reviewer #2: Yes

Reviewer #3: Yes

6. Review Comments to the Author

Reviewer #2: Unfortunately, this Reviewer is not satisfied with the revision provided by the authors.

The nomenclature used throughout the paper is still unacceptable and if the authors are not willing to comply with the nomenclature provided by the Scientific and Standardization Committee of the ISTH (Kohler et al, JTH 2011;9:1404–6), I strongly believe that they should not submit and try to publish in Plos One. I think that this a major point as by not using the correct nomenclature, confusion is generated within the scientific community and the paper will not advance this field. Moreover, my other points (e.g. omission of statistically underpowered analysis) were not taken fully into consideration, and only minor changes (e.g. adding a limitation section) were introduced in the paper. I believe that the paper in its current form is misleading both due to the nomenclature used but also due to contents that are not supported by sound statistical analysis and thus, unfortunately, I cannot agree to the publication of this manuscript in its current form.

Reviewer #3: comments:

(1) Please could the authors provide the information about the antigen epitope of anti-FXIII/13 antibody(from Prof.Reed's gift).

(2) For the quantification of FXIII/13 inhibitory titer,the healthy individuals controls should be included.

(3)Is it only limited in FXIII/13 A chain for the detection of against FXIII/13 autoantibodies.Please could the authors comment or illustrate on why autoantibody againist FXIII/13 is not involved in FXIII/13 B chains.

(4)Please could the authors supplement the diagnostic criteria on acquired autoimmune FXIII/13 deficiency or reference in your study series.

(5)In your present studies,you used two methods(ELISA & IST) for evaluating FXIII/13 autoantibodies levels.Could you please determine which method is more sensitive and specific in identifying autoantibodies against FXIII/13;How about the correlation of the both methods.

7. PLOS authors have the option to publish the peer review history of their article (what does this mean?). If published, this will include your full peer review and any attached files.

Reviewer #2: No

Reviewer #3: No

---

## [Author Response · Author response to Decision Letter 1]

27 Aug 2021

August 27, 2021

Dear Dr. Arijit Biswas:

We appreciate reviewer’s comments very much, for they help us to improve the quality of our paper considerably. According to the reviewer’s comments, the manuscript has been revised with yellow backgrounds and red letters.

Before responding point-by-point, we would like to draw your attention to two major additional changes that were revised according to the reviewers’ point out. First, we changed the abbreviations to comply with the nomenclature provided by the Scientific and Standardization Committee of the ISTH. Second, we accept the Reviewer # 2's criticism concerning statistically underpowered analysis, and significantly rewritten “Association of selected candidate alleles with FXIII inhibitory titers and/or levels of anti-FXIII-A autoantibodies” in the “Results” section.

Response to Reviewer #2

Comment 1–1: The nomenclature used throughout the paper is still unacceptable and if the authors are not willing to comply with the nomenclature provided by the Scientific and Standardization Committee of the ISTH (Kohler et al, JTH 2011;9:1404–6), I strongly believe that they should not submit and try to publish in Plos One.

Reply to Comment 1–1: Following the comments of Reviewer #2, we changed the abbreviations to comply with the nomenclature provided by the Scientific and Standardization Committee of the ISTH (Kohler et al, JTH 2011;9:1404–6). Specifically, FXIII/13 was changed to FXIII (94 places including p. 2, lane 3). In addition, AiF13D and AiF8D were changed to autoimmune FXIII deficiency (74 places including p. 2, lane 2) and AHA (9 places including p.7, lane 15).

Comment 1–2: I think that this a major point as by not using the correct nomenclature, confusion is generated within the scientific community and the paper will not advance this field.

Reply to Comment 1–2: Reviewer # 2's comments are justified. So, to avoid confusion in the scientific community, we changed the abbreviations according to the correct nomenclature as described in “Reply to Comment 1–1”.

Comment 2: Moreover, my other points (e.g. omission of statistically underpowered analysis) were not taken fully into consideration, and only minor changes (e.g. adding a limitation section) were introduced in the paper.

Reply to Comment 2: We accept the Reviewer # 2's criticism concerning statistically underpowered analysis, and have deleted the P-value in Tables 3–5, S9 Table, and Fig. 2. In addition, we made the expression a little less than the assertive tone in “Abstract” (p. 2, lanes 12–17), “Results” (p. 23, lane 15–p. 24, lane 9) and “Discussion” (p. 28, lanes 5–11). The relevant parts of the rest of the manuscript have been altered accordingly (p. 14, lane 1 & legends of Tables 3–5 and S9 Table).

Comment 3: I believe that the paper in its current form is misleading both due to the nomenclature used but also due to contents that are not supported by sound statistical analysis and thus, unfortunately, I cannot agree to the publication of this manuscript in its current form.

Reply to Comment 3: Following the comments of Reviewer #2, we changed the abbreviations to comply with the nomenclature provided by the Scientific and Standardization Committee of the ISTH. In addition, as described in “Reply to Comment 2”, we accept the Reviewer # 2's criticism concerning statistically underpowered analysis, and have deleted the P-value in Tables 3–5, S9 Table, and Fig. 2 to avoid misleading. We believe that the revised manuscript is now suitable for publication.

Response to Reviewer #3

Comment 1: Please could the authors provide the information about the antigen epitope of anti-FXIII/13 antibody (from Prof. Reed's gift).

Reply to Comment 1: Unfortunately, we do not have information about the antigen epitope of the anti-FXIII antibody. However, since this antibody recognizes FXIIIa and FXIII-A2B2, it is considered that it does not recognize the activation peptide or the binding site with FXIII-B.

Comment 2: For the quantification of FXIII/13 inhibitory titer, the healthy individual controls should be included.

Reply to Comment 2: I would like to include data for healthy individual controls in order to increase the number, but the inhibitory titer of healthy individual controls cannot be calculated. One Bethesda unit is defined as the amount of inhibitor in 1 mL of plasma that will neutralize 50% of the FXIII activity (residual activity = 50%). The numbers of Bethesda units are calculated according to the dilution of the patient's plasma. To determine the BU/ mL, it is necessary to obtain the residual activity between 25 and 75%. However, FXIII activity of healthy individual controls is generally >70%, and the FXIII activity of 1:1 mixture of standard plasma and healthy individual control plasma is >75%. Therefore, we cannot calculate FXIII inhibitory titer of healthy individual controls.

Comment 3: Is it only limited in FXIII/13 A chain for the detection of against FXIII/13 autoantibodies. Please could the authors comment or illustrate on why autoantibody against FXIII/13 is not involved in FXIII/13 B chains.

Reply to Comment 3: Ninety-five percent of the cases with the autoimmune FXIII deficiency we have identified were due to autoantibodies against FXIII-A and the remaining 5% were due to autoantibodies against FXIII-B. As described in S1 Table, case No. 48 used in this study was due to autoantibodies against FXIII-B. However, in these cases, the decrease in the level of FXIII antigen due to clearance is the cause of the decrease in activity, and since there is no neutralizing activity, there is almost no inhibitory titer.

Comment 4: Please could the authors supplement the diagnostic criteria on acquired autoimmune FXIII/13 deficiency or reference in your study series.

Reply to Comment 4: All patients with acquired autoimmune FXIII deficiency met the diagnostic criteria from the ISTH/SSC FXIII/Fibrinogen Subcommittee (Ichinose A et al, Thromb Haemost 2016; 116: 772–774). Following the comments of Reviewer #3, we supplemented the reference in the “Clinical samples” section of the “Materials and methods” section (p. 7, lane 7).

Comment 5: In your present studies, you used two methods (ELISA & IST) for evaluating FXIII/13 autoantibodies levels. Could you please determine which method is more sensitive and specific in identifying autoantibodies against FXIII/13; How about the correlation of the both methods.

Reply to Comment 5: As mentioned in the “Discussion” section (p. 31, lane 5–p. 32, lane 2), we believe that the ELISA method is more specific in identifying autoantibodies against FXIII-A. In fact, the logarithm of the FXIII inhibitory titer was better correlated with the ELISA method (R=0.86, P<0.0001) than with the ICT method (R=0.63, P=0.0037). However, for patients with many low-affinity antibodies to FXIII-A, the ICT method may be more sensitive. The correlation between the two methods was moderately good (R=0.61, P=0.0043).

---

## [Editor Report · Decision Letter 2]

31 Aug 2021

Important roles of the human leukocyte antigen class I and II molecules and their associated genes in the autoimmune coagulation factor XIII deficiency via whole-exome sequencing analysis

PONE-D-21-03903R2

Dear Dr. Osaki,

We’re pleased to inform you that your manuscript has been judged scientifically suitable for publication and will be formally accepted for publication once it meets all outstanding technical requirements.

Kind regards,

Arijit Biswas

Academic Editor

PLOS ONE
---

## [Editor Report · Acceptance letter]

2 Sep 2021

PONE-D-21-03903R2 

Important roles of the human leukocyte antigen class I and II molecules and their associated genes in the autoimmune coagulation factor XIII deficiency via whole-exome sequencing analysis 

Dear Dr. Osaki:

I'm pleased to inform you that your manuscript has been deemed suitable for publication in PLOS ONE. Congratulations! Your manuscript is now with our production department. 

Kind regards, 

on behalf of

Dr. Arijit Biswas 

Academic Editor

PLOS ONE